# LASSO: LATENT SUB-SPACES ORIENTATION FOR DOMAIN GENERALIZATION

## ABSTRACT

To achieve a satisfactory generalization performance on prediction tasks in an unseen domain, existing domain generalization (DG) approaches often rely on the assumption of the existence of fixed domain-invariant features and common hypotheses learned from a set of training domains. While it is a natural and important premise to ground generalization capacity on the target domain, we argue that this assumption could be overly strict and sub-optimal. It is particularly evident when source domains share little information or the target domains leverages information from selective source domains in a compositional way instead of relying on a unique invariant hypothesis across all source domains. Unlike most existing approaches, instead of constructing a single hypothesis shared among domains, we propose a LAtent Sub-Space Orientation (LASSO) method that explores diverse latent sub-spaces and learning individual hypotheses on those sub-spaces. Moreover, in LASSO, since the latent sub-spaces are formed by the label-informative features captured in source domains, they allow us to project target examples onto appropriate sub-spaces while preserving crucial label-informative features for the label prediction task. Finally, we empirically evaluate our method on several well-known DG benchmarks, where it achieves state-of-the-art results.

## 1 INTRODUCTION

One of the most challenging problems in applying machine learning to real-world problems is to address the domain shift encountered when test data at inference time come from different distributions compared to training data, often causing unexpectedly imperfect generalization performance. To handle this issue, many out-of-distribution learning settings have been investigated, notably domain adaptation (DA) and domain generalization (DG). In particular, DA setting (Mansour et al., 2009; Ben-David et al., 2010; Zhao et al., 2019) takes the assumption that both labeled source data and unlabeled target data are available at the training phase, while DG setting (Blanchard et al., 2011; Muandet et al., 2013; Ganin et al., 2016) is much more challenging due to the complete absence of any target data at training time, and the learned model is expected to perform a zero-shot prediction on test samples. While being more challenging than DA, DG is arguably more versatile and applicable to real-world scenarios where there is a need to rapidly deploy a prediction model on a new target domain without any access to target data.

A common premise on which most of the existing DG approaches rely to address the domain shift problem is to learn a shared set of features among different source domains, e.g., domain-invariant features (Muandet et al., 2013; Ganin et al., 2016; Motiian et al., 2017; Ghifary et al., 2015; Xie et al., 2017; Wang et al., 2019; Piratla et al., 2020; Zhao et al., 2020). By learning domain-invariant features, these approaches identify a latent representation from multiple source domains which is also expected encompass unseen target domains, hence reducing the domain shift. However, since this latent space is often high-dimensional and unseen target domains could vary greatly and unexpectedly, these domain invariance approaches could become ineffective. We argue that using a single hypothesis which requires to be optimal for all source and target domains is rather a strong assumption and might not have an effective algorithmic solution, hence presenting a key weakness to these existing approaches.

In this paper, we propose a new approach to efficiently reduce the domain shift. Our idea is to optimally decompose the high-dimensional latent feature space to lower-dimensional sub-spaces,

after which an individual hypothesis is learned (optimally) for each latent sub-space. In particular, these latent sub-spaces are formed by label-informative features which are learned from source latent representations so that sub-space latent representations can capture sufficient and necessary information to effectively train sub-space hypotheses. We then perform extensive experimental evaluations to demonstrate the merits of our methodology and framework. In sum, our contributions are as follows:

- We propose LAtent SubSpace Orientation (LASSO), a novel approach for DG, which aims to learn label-informative latent sub-spaces from source latent representations, preserving crucial label information for training qualified individual sub-space hypotheses.

- We develop a rigorous theoretical analysis to explain how our approach can learn meaningful sub-spaces by learning an indicator function $\Gamma$ that helps to reduce latent domain shift. Moreover, by linking to information theory (cf. Theorem 3), we theoretically justify the rationale behind learning the sub-space indicator function $\Gamma$ as a mechanism for element-wise selection of latent features to gather sufficient label information.

- Finally, we empirically demonstrate that our proposed method can achieve favorable results when evaluating our model on domain generalization benchmarks in the comparison with state-of-the-art methods.

## 2 RELATED WORK

Domain generalization (DG) approaches can be categorized into several groups: *domain-invariant representation learning*, *meta-learning*, and *augmentation/self-supervision*. The works in the first group aim to learn a domain-invariant representation with the hope of transferring well to unseen domains. Notably, Muandet et al. (2013) construct shared components by minimizing the discrepancy of the source domain marginal distributions using a kernel-based algorithm. Xie et al. (2017) employ an adversarial training strategy to preserve the desired invariance and eliminate variations of domain factors from features. Seo et al. (2020) combine batch normalization and instance normalization to remove domain-specific styles while preserving semantic category information. Other works (Ghifary et al., 2015; Li et al., 2018b; Ilse et al., 2020) employ auto-encoders to support domain-invariant feature extraction.

Another efficient approach for DG is meta-learning. For example, the studies in (Li et al., 2018a; Balaji et al., 2018) use source domains to simulate meta-train/meta-test, which encourages the trained model to generalize better on the meta-test data, leading to a more plausible global generalization performance on the unseen target domain. Dou et al. (2019) then extend the work of Li et al. (2018a) by combining it with metric learning loss to encourage domain-independent semantic feature space.

Self-supervised learning and data augmentation have been also applied to DG. Typically, Carlucci et al. (2019) propose to solve the pretext task of Jigsaw Puzzles to improve the generalization performance on unseen domains. Shankar et al. (2018) augment training data with instances perturbed along with directions of domain change. In addition, Zhou et al. (2020a) employ a classifier that can learn the generalization on additional augmented samples of diversity pseudo-novel domains by leveraging optimal transport theory. Zhou et al. (2020b) augment the original training data of source domains with synthetic data from unseen domains such that augmented data are correctly classified by the classifier while fooling the domain classifier to make the task model intrinsically more domain-generalizable. Zhou et al. (2021) propose to mix the styles of different source domains based on normalization-based style-transfer technique to effectively increase the diversity of domains during training.

Recently, learning domain-specific information to boost classification performance in domain generalization has attracted more attention. For example, Huang et al. (2020) iteratively discard the dominant features to exploit all useful features (including both invariant and specific features) that highly correlate with labels. Chattopadhyay et al. (2020) propose a domain-specific mask learned from the domain label to balance domain-invariant and domain-specific features. As a result, source domain classification can benefit from the specialized features while retaining the generalizing applicability of domain-invariant features.

## 3 PROPOSED METHOD

### 3.1 PROBLEM SETTING

Let $\mathbb{D}_k^S = \{(x_i^k, y_i^k)\}_{i=1}^{N_k^S}$, $k = 1, ..., K$ be the source labeled datasets, where $k$ is the domain index, $N_k^S$ is the number of examples in $\mathbb{D}_k^S$ and $y_i^k \in \mathcal{Y} := \{1, ..., C\}$ is the set of $C$ classes. Let $\mathbb{P}_k^S$ and $p_k^S$ be the (marginal) data distribution and corresponding density of the source domain $k$. Similarly, let $p_k^S(y \mid x)$ denote the probabilistic labeling distribution of this domain. A domain is then defined by a pair of data distribution and labeling function, in which its data $(x_i^k, y_i^k)$ is generated by first sampling $x_i^k \sim p_k^S(\cdot)$ and subsequently $y_i^k \sim p_k^S(\cdot \mid x_i^k)$. With a little abuse of notation, let us also denote $k$-th domain by $\mathbb{D}_k^S$, and the data generation process is $(x, y) \sim \mathbb{D}_k^S$. For target domain $\mathbb{D}^T$, $\mathbb{P}^T$ and $p^T$ are its data distribution and density respectively, while $p^T(y \mid x)$ is the target labeling distribution.

A mixture of the source domains is denoted by $\mathbb{D}_\pi^S := \sum_{k=1}^K \pi_k \mathbb{D}_k^S$. In order to sample data from this mixture, one first sample a domain index from the categorical distribution $k \sim \text{Cat}(\pi)$, where $\pi$ specifies the mixture weights, and then sample the actual data from the corresponding domain $k$. The empirical datasets obtained could hint us to estimate $\pi_k = \left[\frac{N_k^S}{N^S}\right]_{k=1}^K$ with $N^S = \sum_{k=1}^K N_k^S$. Finally, a mixture of data distribution is defined by $\mathbb{P}_\pi^S = \sum_{k=1}^K \pi_k \mathbb{P}_k^S$.

A hypothesis $f : \mathcal{X} \to \Delta_C$ is a map from the common data space $\mathcal{X}$ to the the $C$-simplex label space $\Delta_C := \{\alpha \in \mathbb{R}^C : \|\alpha\|_1 = 1 \wedge \alpha \geq 0\}$. Let $\ell(f(x), y)$ be the loss incurred by using this hypothesis to predict $x \in \mathcal{X}$, given its ground-truth label $y \in \mathcal{Y}$. Therefore, the general loss of the hypothesis $f$ w.r.t. the joint distribution $\mathbb{D}$ is:

$$\mathcal{L}(f, \mathbb{D}) = \mathbb{E}_{(x,y) \sim \mathbb{D}}[\ell(f(x), y)]. \tag{1}$$

We examine the composite hypothesis $f = h \circ g$ where $g : \mathcal{X} \to \mathcal{Z}$ is the feature extractor mapping the data space to a latent space and $h : \mathcal{Z} \to \mathcal{Y}$ is the classifier on this latent space. With respect to the latent space, given a probabilistic labeling distribution $p(y \mid x)$, let denote $p^g(y \mid z)$ as the probabilistic labeling distribution induced by $p(y \mid x)$ and $g$, that is, for any $z \in g(\mathcal{X})$, $p^g(y \mid z) = \frac{\int_{g^{-1}(z)} p(y|x)p(x)dx}{\int_{g^{-1}(z)} p(x)dx}$ (Johansson et al., 2019). Basically, $p^g(y \mid z)$ can be regarded as a weighted sum of $p(y \mid x)$ for $x \in g^{-1}(z)$. We further define $p_k^{S,g}(y = c \mid z)$ and $p^{T,g}(y = c \mid z)$ with $z \in g(\mathcal{X}) \subset \mathcal{Z}$ as the induced conditional labeling distributions of source and target domains on the latent space via the feature extractor $g$. Finally, we consider the DG setting, in which we only possess the source labeled data and do not have any prior information of any target data during the training process.

In what follows, we develop a rigorous theory to promote our LASSO, whose proofs can be found in the Appendix A.

### 3.2 SINGLE HYPOTHESIS AND FULL SPACE FOR DOMAIN GENERALIZATION

Most existing works in DG use a single hypothesis acting on the entire latent space $\mathcal{Z}$ for predicting unseen target domains. In what follows, we develop an upper-bound which is relevant to the data shift on the latent space of the target loss $\mathcal{L}(f, \mathbb{D}^T)$ with $f = h \circ g$.

**Theorem 1.** *If the loss function $\ell$ is upper-bounded by a positive constant $L$, for any hypothesis $f : \mathcal{X} \to \Delta_C$ where $f = h \circ g$ with $g : \mathcal{X} \to \mathcal{Z}$ and $h : \mathcal{Z} \to \Delta_C$, the target general loss is upper-bounded by:*

$$\mathcal{L}(f, \mathbb{D}^T) \leq \exp\left\{R^\alpha\left(g_\# \mathbb{P}^T, g_\# \mathbb{P}_\pi^S\right)\right\}^{\frac{\alpha-1}{\alpha}} L^{\frac{1}{\alpha}} \left[\mathcal{L}(f, \mathbb{D}_\pi^S) + L \max_k \mathbb{E}_{\mathbb{P}_k^S}[\|\Delta p_k(y \mid x)\|_1]\right]^{\frac{\alpha-1}{\alpha}},$$
$$\tag{2}$$

*where $g_\# \mathbb{P}_\pi^S$ and $g_\# \mathbb{P}^T$ are pushed-forward distributions induced by applying $g$ on $\mathbb{P}_\pi^S$ and $\mathbb{P}^T$, $R^\alpha$ is the $\alpha$-divergence ($\alpha > 1$), and $\Delta p_k(y \mid x) := \left[\left|p_k^S(y = c \mid x) - p^T(y = c \mid x)\right|\right]_{c=1}^C$ represents the label shift between the labeling assignment mechanisms of an individual source domain and target domain on the input space.*

The bound developed in Theorem 1 supports us in characterizing the factors influencing the loss on the target domain of a hypothesis $f$: (i) *the label shift*: $\mathbb{E}_{\mathbb{P}_k^S}\left[\|\Delta p_k\left(y \mid x\right)\|_1\right]$, (ii) *the source loss*: $\mathcal{L}\left(f, \mathbb{D}_\pi^S\right)$, and (iii) *the latent data shift*: $R^\alpha\left(g_\#\mathbb{P}^T, g_\#\mathbb{P}_\pi^S\right)$. Based on this theorem, we aim to find the hypothesis $f$ such that the upper bound of target loss is minimized. Note that the *label shift term* is a natural characteristic of domains, hence it is almost unchangeable. Therefore, one seeks to minimize the source loss, or the discrepancy between latent distributions, or both of them. Certainly, if we know target data (e.g., samples from $\mathbb{P}^T$), we can minimize the divergence between $g_\#\mathbb{P}_\pi^S$ and $g_\#\mathbb{P}^T$ directly. However, this is almost impossible for the DG setting because the target domain is unknown beforehand when training.

To address this problem, a large number of works propose learning domain-invariant features on a full high-dimensional latent space together with a single hypothesis on top of these domain-invariant features. Nonetheless, due to the great variance of unseen target distributions on the full high-dimensional latent space, the latent data shift is possibly high in many cases, which hurts the generalization ability of the single hypothesis on unseen target domains.

Evidently, given a latent representation $z$ with a label $y$ in the high-dimensional latent space, only a small portion of its features known as *label-informative features* is highly relevant to the label $y$, while the remaining ones are redundant. By eliminating irrelevant features and grouping the latent representations of data examples across multiple domains with the same set of label-informative features, we can form latent label-informative sub-spaces to reduce the latent data shift, whereas preserving *sufficient label-information* for training *good hypotheses* on those *latent sub-spaces*. This can be explained from the fact that unseen target and source examples with the same label-informative features are projected onto the same latent sub-space on which the latent data shift between mixture of source domains and target domain becomes smaller due to the compactness of this sub-space compared to the full high-dimensional latent space.

Furthermore, by learning multiple hypotheses, each of which corresponds to a sub-space, LASSO allows sub-spaces to instance-wisely explore different groups of label-informative features, hence encouraging the diversity of latent representations for achieving better generalization ability, which concurs with the principle in (Huang et al., 2020; Chattopadhyay et al., 2020; Blanchard et al., 2021). This intuitively boosts the generalization ability of hypotheses. The reason is that given source examples with their labels, LASSO aims to explore possible compact groups of label-informative features which can predict accurately the labels. In the inference time, given a target example, if the feature extractor can successfully activate a group of label-informative features, this target example is projected and matched with corresponding source examples on a sub-space in which a good hypothesis is used to predict a label for the target example.

### 3.3 LATENT SUB-SPACES WITH INDIVIDUAL HYPOTHESES FOR DOMAIN GENERALIZATION

For a data sample $(x, y) \sim \mathbb{D}_\pi^S$, we consider latent representation $z = g(x) \in \mathbb{R}^D$ and propose to learn a *sub-space indicator* $\Gamma$ which renders $\Gamma(z) \in \{0,1\}^D$. Specifically, $\Gamma(z)$ specifies the *sub-space* which we project $z$ onto via $\Pi_\Gamma(z) = z \odot \Gamma(z)$, where $\odot$ is the element-wise product operator. Our intuition is that $\Gamma$ is able to keep the most label informative features of latent vector $z$ (i.e., the $d^{\text{th}}$ element of $\Gamma(z)$: $\Gamma_d(z) = 1$ means that $z_d$ is label-informative and should be selected when projecting $z$ onto its sub-space).

Given a sub-space index $m \in \{0,1\}^D$, we denote the region on data space which has the same index $m$ as $A_m = \{x : \Gamma(g(x)) = m\} \subset \mathcal{X}$. Let $\mathbb{P}_m^S$ be the distribution restricted by $\mathbb{P}_\pi^S$ over the set $A_m$ and $\mathbb{P}_m^T$ as the distribution restricted by $\mathbb{P}^T$ over $A_m$. Eventually, we define $p_m^S\left(y \mid x\right)$ as the probabilistic labeling distribution on the sub-space $\left(A_m, \mathbb{P}_m^S\right)$, meaning that if $x \sim \mathbb{P}_m^S$, $p_m^S\left(y \mid x\right) = \sum_{k=1}^K \pi_k p_k^S\left(y \mid x\right)$. Similarly, we define if $x \sim \mathbb{P}_m^T$, $p_m^T\left(y \mid x\right) = p^T\left(y \mid x\right)$.

Due to this construction, any data sampled from $\mathbb{P}_m^S$ or $\mathbb{P}_m^T$ have the same index $m = \Gamma(z) = \Gamma(g(x))$, which specifies the set of relevant feature elements for classifying data from $A_m$. This hints us to employ an individual hypothesis $f_m = h_m \circ g^\Gamma$, where $f_m$ specializes on $\left(A_m, \mathbb{P}_m^S\right)$ and $h_m$ specializes on the sub-space $\left(g^\Gamma\left(A_m\right), g^\Gamma_\#\mathbb{P}_m^S\right)$ with $g^\Gamma := \Pi_\Gamma \circ g$ for each $m$. We note that $h_m$ operates on the sub-space indexed by $m$, while $f_m$ operates on its induced data space. Additionally, since each data point $x \in \mathcal{X}$ corresponds to only a single $\Gamma(g(x))$, the data space is partitioned

into disjoint sets, i.e., $\mathcal{X} = \bigcup_{m=1}^{\mathcal{M}} A_m$, where $A_m \cap A_n = \emptyset, \forall m \neq n$. Therefore, these different hypotheses $h_m$ make use of different features, as reflected in the different sub-space index $m$.

We now present the way to efficiently formulate sub-space hypotheses. Let $\mathcal{F} := \{f = h \circ g : g \in \mathcal{G} \wedge h \in \mathcal{H}\}$ be a hypothesis class of $f : \mathcal{X} \rightarrow \Delta_C$, where $f = h \circ g$ with $g : \mathcal{X} \rightarrow \mathcal{Z} \in \mathcal{G}$ and $h : \mathcal{Z} \rightarrow \Delta_C \in \mathcal{H}$. Each hypothesis $f = h \circ g$ induces the sub-space hypotheses $f_{\mathcal{M}} = [f_m]_{m \in \mathcal{M}}$ for which $f_m(x) = h_m(\Gamma(g(x)) \odot g(x))$ for $x \in A_m$ with $h_m(m \odot g(x)) = h(m \odot g(x))$ can be viewed as a hypothesis on the sub-space indexed by $m$. We define the general loss of $f_{\mathcal{M}} = [f_m]_{m \in \mathcal{M}}$ on the target domain as

$$\mathcal{L}(f_{\mathcal{M}}, \mathbb{D}^T) := \frac{1}{|\mathcal{M}|} \sum_{m \in \mathcal{M}} \mathcal{L}(f_m, \mathbb{D}_m^T), \tag{3}$$

where $\mathcal{M} \subset \{0,1\}^D$ is the set of all feasible sub-space indices $m$ conducted from the observed data, $|\mathcal{M}|$ specifies the cardinality of $\mathcal{M}$, and $\mathbb{D}_m^T$ is the joint distribution of $(x, y)$ where $x \sim \mathbb{P}_m^T$ and $y \sim p_m^T(\cdot \mid x)$.

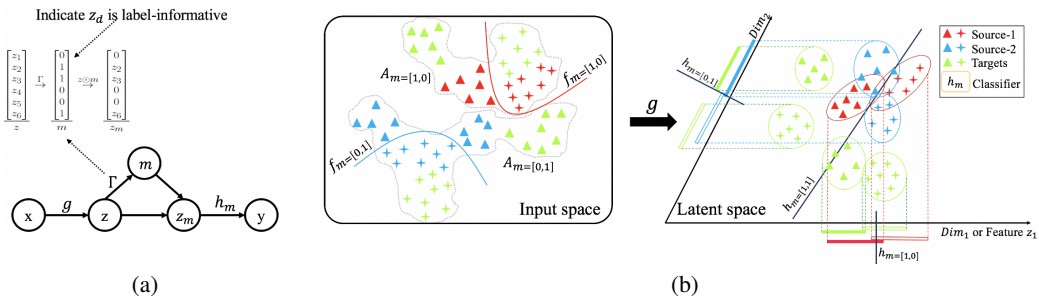

(a)              (b)

Figure 1: (a) Latent Sub-space Orientation Framework and (b) An example of reducing data shift between source domains and target domain in the latent sub-spaces.

Figure 1a depicts the generative process of our LASSO in which the sub-space indicator $\Gamma$ specifies the sub-space to project $z = g(x)$ onto and the projected sample is then predicted by a sub-space hypothesis. In Figure 1b, we intuitively demonstrate why projecting onto appropriate sub-spaces is useful to reduce latent data shift. For the red points on the latent space, the dimension 1 is commonly label-informative, hence projecting onto this dimension is helpful to preserve label information for constructing a good hypothesis on this sub-space. The same observation occurs for the blue points for the dimension 2. The green points symbolize target data for which we observe that the latent data shift on the full latent space is significantly greater than that on the sub-spaces. In addition, Figure 1b also represents some notions in our developed theory, i.e., $A_{m=[1,0]}$ with the distribution $\mathbb{P}_{m=[1,0]}$ includes the data samples whose latent representations are projected onto the sub-space indexed by $m = [1,0]$ and similar notions for another sub-space. Additionally, each sub-space index $m$ associates with two equivalent hypotheses $f_m$ on $A_m$ and $h_m$ on the sub-space $m$.

The following theorems explain the intuition that matching source and target domains on good sub-spaces helps reduce the latent data shift w.r.t. a general unseen target domain; and offer us the training process of the *sub-space indicator* $\Gamma$, which extracts the most label informative features of latent representations $z$.

**Theorem 2.** *Given a sub-space indicator $\Gamma$, if the loss function $\ell$ is upper-bounded by a positive constant $L$, the sub-space target general loss is upper-bounded by:*

$$\mathcal{L}(f_{\mathcal{M}}, \mathbb{D}^T) \leq \frac{L^{\frac{1}{\alpha}}}{|\mathcal{M}|} \sum_{m \in \mathcal{M}} \exp\left\{ R^\alpha \left( g^\Gamma_{\#} \mathbb{P}_m^T, g^\Gamma_{\#} \mathbb{P}_m^S \right) \right\}^{\frac{\alpha-1}{\alpha}} \left[ \mathcal{L}(f_m, \mathbb{D}_m^S) + L\Delta p_m \right]^{\frac{\alpha-1}{\alpha}}, \tag{4}$$

*where $\mathbb{D}_m^S$ is the joint distribution of $(x, y)$ with $x \sim \mathbb{P}_m^S$ and $y \sim p_m^S(\cdot \mid x)$, and $\Delta p_m := \mathbb{E}_{x \sim \mathbb{P}_m^S}\left[ \left\| p_m^S(\cdot \mid x) - p_m^T(\cdot \mid x) \right\|_1 \right]$.*

Theorem 2 can be viewed as an extension of Theorem 1 in the context of LASSO with multiple sub-spaces. This theorem shows an upper-bound of the sub-space target loss which consists of three terms: (i) *the sub-space label shift*: $\Delta p_m$, (ii) *the source sub-space hypothesis losses*: $\mathcal{L}(f_m, \mathbb{D}_m^S)$, and (iii) *the sub-space latent data shift*: $R^\alpha\left( g^\Gamma_{\#} \mathbb{P}_m^T, g^\Gamma_{\#} \mathbb{P}_m^S \right)$.

Since the sub-space label shift depends on the natural characteristics of domains and the source sub-space hypothesis losses are trainable, the *sub-space latent data shift* (i.e., the divergence between the mixture of source domains: $g^{\Gamma}{}_{\#}\mathbb{P}_m^S$ and target domain: $g^{\Gamma}{}_{\#}\mathbb{P}_m^T$ on a sub-space) is essential to lower the upper-bound in (4). As demonstrated in Figure 1b, by projecting corresponding latent representations onto low-dimensional sub-spaces conducted from label-informative features, we can reduce the latent data shift on the sub-spaces due to the compression effect when projecting to the sub-spaces. Moreover, if we can appropriately choose the good sets of label-informative features for the sub-spaces, we can preserve sufficient label information on the sub-spaces for training good hypotheses $f_m$ or $h_m$ with a low source sub-space hypothesis loss $\mathcal{L}\left(f_m, \mathbb{D}_m^S\right)$ with the aim to lower the upper-bound in (4).

The next arising question regarding how to train a qualified *sub-space indicator* $\Gamma$ for choosing good sets of label-informative features to conduct low-dimensional and label-preserving sub-spaces is addressed in the following theorem.

**Theorem 3.** *Let $X$ is a random variable of source sample (i.e., drawn from $\mathbb{P}_\pi^S$) and $Y$ is a random variable of ground-truth labels. Denote $N = \sum_{m' \in \mathcal{M}} \mathbb{P}_\pi^S\left(A_{m'}\right)$, we then have*

$$\mathbb{I}\left(\Gamma\left(g(X)\right) \odot g\left(X\right), Y\right) \geq - \sum_{m \in \mathcal{M}} \frac{\mathbb{P}_\pi^S\left(A_m\right)}{N} \mathcal{L}\left(f_m, \mathbb{D}_m^S\right) + const, \tag{5}$$

*where $\mathcal{L}\left(f_m, \mathbb{D}_m^S\right)$ is defined based on the cross-entropy loss and $\mathbb{I}$ denotes mutual information.*

Theorem 3 gives us a hint of how to train a *sub-space indicator* $\Gamma$ together with the *sub-space hypotheses* $f_\mathcal{M} = [f_m]_{m \in \mathcal{M}}$. As suggested by Theorem 3, we minimize the *source sub-space hypothesis* losses $\mathcal{L}\left(f_m, \mathbb{D}_m^S\right)$ by

$$\min_{\Gamma, g, h} \sum_{m \in \mathcal{M}} \frac{\mathbb{P}_\pi^S\left(A_m\right)}{N} \mathcal{L}\left(f_m, \mathbb{D}_m^S\right) = \min_{\Gamma, g, h} \sum_{m \in \mathcal{M}} \frac{\mathbb{P}_\pi^S\left(A_m\right)}{N} \mathbb{E}_{\mathbb{D}_m^S}\left[\ell\left(h\left(m \odot g\left(x\right)\right), y\right)\right]$$

$$= \min_{\Gamma, g, h} \mathbb{E}_{\mathbb{D}_\pi^S}\left[\ell\left(h\left(m \odot g\left(x\right)\right), y\right)\right]. \tag{6}$$

Furthermore, Theorem 3 indicates that by solving the optimization problem in (6), we implicitly learn $\Gamma, g$ to maximize the mutual information $\mathbb{I}\left(\Gamma\left(g(X)\right) \odot g\left(X\right), Y\right)$, which enables the *sub-space indicator* $\Gamma$ to preserve the label-informative features of $z = g\left(x\right)$.

## 3.4 LATENT SUB-SPACES LEARNING FRAMEWORK

In what follows, we present the technical details of our proposed method which solves the optimization problem in equation 6. In particular, we discuss the way to formulate and learn the sub-space indicator $\Gamma$, the feature extractor $g : \mathcal{X} \to \mathcal{Z}$, and the classifier $h : \mathcal{Z} \to \Delta_C$, in which $\Gamma$ and $(g, h)$ are updated alternatively.

### 3.4.1 SUB-SPACE INDICATOR

We employ a probabilistic *sub-space indicator* $\Gamma\left(z\right) \in [0, 1]^D$ in which $\Gamma_d\left(z\right) \in [0, 1]$ represents the probability that $z_d$ is a label-informative feature. Additionally, in our implementation, we consider a 2D tensor $z \in \mathbb{R}^{D' \times D}$, whereas each $z_d \in \mathbb{R}^{D'}$ known as *an attribute* consists of $D'$ features which are simultaneously selected or unselected as the dimensions of the projection sub-space. The probabilistic sub-space indicator is formulated as $\Gamma\left(z\right) = \left[\Gamma_1\left(z_1\right), ..., \Gamma_D\left(z_D\right)\right] \in [0, 1]^D$. The element-wise product $\Gamma\left(z\right) \odot z$ is defined as $z\mathrm{diag}\left(\Gamma\left(z\right)\right)$, wherein we multiply each element in $\Gamma\left(z\right)$ to a column of $z$ to select or unselect the group of features in this column. Consequently, $\Gamma_d\left(z\right) = \Gamma_d\left(z_d\right)$ is computed based solely on the group of $D'$ features in the attribute $z_d$ itself rather than full latent $z$. That means attribute-based $\Gamma_d$ depends on attributes which are shared across domains instead of domains, hence, becomes more independent from domain information. Moreover, by Theorem 3, model can still learn meaningful attributes as long as the performance on source domains is guaranteed. In unseen domains, the target of sub-space indicator $\Gamma$ is to detect label-informative attributes *which are learned in source domains* instead of identifying all label-informative attributes. For further discussion about attribute-based $\Gamma_d$ please refer to Appendix B.1. Finally, we minimize the following loss for updating $\Gamma$:

$$\mathcal{L}_I\left(\Gamma | g, h\right) = \mathbb{E}_{(x,y) \sim \mathbb{D}_\pi^S}\left[\ell\left(h\left(\Gamma\left(z\right) \odot z\right), y\right)\right]_{|z = g(x)}. \tag{7}$$

### 3.4.2 Sub-space hypotheses

To encourage the sub-space exploration and strengthen the sub-space hypotheses, given a data-label pair $(x, y) \sim \mathbb{D}_\pi^S$, we sample mask indicators from the Berboulli distribution with parameter $\Gamma_d(z_d)$, i.e,. $m_d \sim Ber(\Gamma_d(z_d))$ to gather $m = [m_d]_{d=1}^D$, and then project onto a sub-space by $m \odot z$. Then, $g, h$ are updated by minimizing the following loss function:

$$\mathcal{L}_H(g, h | \Gamma) = \mathbb{E}_{(x,y) \sim \mathbb{D}_\pi^S} \left[ \mathbb{E}_{m \sim Ber(\Gamma(z))} \left[ \ell(h(m \odot z), y) \right] \right]_{|z=g(x)}, \tag{8}$$

where $m \sim Ber(\Gamma(z))$ means $m_d \sim Ber(\Gamma_d(z_d)), d = 1, ..., D$. For further discussion about mask sampling please refer to Appendix B.2.

Finally, the pseudcode of our LASSO is summarized in Algorithm 1.

---

**Algorithm 1** LASSO: Latent Sub-space Orientation for DG.

---

1: Initialize: encoder $g$, classifier $h$, *Sub-space indicator* $\Gamma$ and dataset $\mathbb{D}_\pi^S$.
2: **for** epoch $= 1 \rightarrow$ epochs **do**
3:     **for** ite **in** iterations **do**
4:         Sample Mini-batch: $\mathbb{B} = \left\{ (x_i, y_i) \sim \mathbb{D}_\pi^S \right\}$
5:         Optimize $\mathcal{L}_I$ in Eq. (7) w.r.t $\Gamma$ on $\mathbb{B}$.
6:         Optimize $\mathcal{L}_H$ in Eq. (8) w.r.t. $h$ and $g$ on $\mathbb{B}$.
7:     **end for**
8: **end for**
9: **Return:** The optimal: $g^*$, $h^*$ and $\Gamma^*$.

---

### 3.4.3 Inference process

At the testing phase, we use two inference strategies: threshold and ensemble strategies

- **Ensemble**: for each target example $x$, we could compute $z = g(x)$, then sample $T$ masks $m_1, ..., m_T \sim Ber(\Gamma(z))$, and ensemble as $h_{\text{En}} = \frac{1}{T} \sum_{t=1}^T h(m_t \odot z)$. However, for efficient computation, we approximate the ensemble prediction by: $h_{\text{En}} \approx h(\Gamma(z) \odot z)$.

- **Threshold:** we employ a threshold $\tau \in [0, 1]$ to compute the mask. Specifically, for each target example $x$, we compute $z = g(x)$, then evaluate the mask $m = \left[ \mathbf{1}_{\Gamma_d(z_d) > \tau} \right]_{d=1}^D$, where $\mathbf{1}$ is the indicator function, and predict as $h(m \odot z)$.

## 4 Experiments

### 4.1 Evaluation on Benchmark Datasets

We report the empirical results on PACS (Li et al., 2017), VLCS (Torralba & Efros, 2011), and Office-Home (Venkateswara et al., 2017) datasets using standard backbones such as AlexNet, ResNet18, and ResNet50 for the feature extractor. Due to the paper space limit, we leave the details of evaluation protocol, experimental settings and implementation in Appendix C.

The results on VLCS using AlexNet are reported in Table 1, the results on Office-Home using ResNet18 and ResNet50 are reported in Table 2 and the results on PACS using three backbones are reported in Table 3 respectively. Those empirical results clearly show that our LASSO provides competitive classification accuracy compared to the baselines. Especially, the average results on Office-Home and PACS in Table 2 and Table 3 consistently show that our proposed method LASSO outperforms the ERM baseline with large margins on both backbones ResNet18 and ResNet50. Although some other baselines also get comparative or better gains than ours with ResNet18, the gains shrink with ResNet50 since larger ResNet backbones are known to generalize better (Gulrajani & Lopez-Paz, 2021). Additionally, the threshold variant LASSO-$\tau$ aims to select and retain the top most relevant and label-informative features (the most appropriate sub-space), while the ensemble variant LASSO-En aggregates the predictions of possible sub-space hypotheses. Therefore, the slight superior of the *threshold* variant to the *ensemble* one is possibly due to the fact that the former with an appropriate $\tau$ can select more compact and label-informative sets of features.

Table 1: Classification Accuracy on VLCS.

| Method | Backbone | PASCAL VOC | LabelMe | Caltech | Sun | Average |
|---|---|---|---|---|---|---|
| ERM (Vapnik, 1999) | AlexNet | 70.58±0.00 | 59.72±0.00 | 96.25±0.00 | 64.51±0.00 | 72.56 |
| MASF (Dou et al., 2019) | AlexNet | 69.14±0.00 | **64.90±0.00** | 94.78±0.00 | **67.64±0.00** | _74.11_ |
| JiGen (Carlucci et al., 2019) | AlexNet | 70.62±0.00 | 60.90±0.00 | 96.93±0.00 | 64.30±0.00 | 73.19 |
| SFA-A (Li et al., 2021) | AlexNet | 70.40±0.00 | 62.00±0.00 | 97.20±0.00 | 66.20±0.00 | 74.00 |
| LASSO-$\tau$ | AlexNet | **71.55±0.21** | 62.42±0.42 | **97.33±0.22** | 65.15±0.22 | **74.15** |
| LASSO-En | AlexNet | _71.37±0.08_ | 62.26±0.19 | _97.17±0.31_ | 65.25±0.26 | 74.01 |

Table 2: Classification Accuracy on Office-Home

| Method | Backbone | Art | Clipart | Product | RealWorld | Average |
|---|---|---|---|---|---|---|
| ERM (Vapnik, 1999) | ResNet18 | 52.15±0.00 | 45.86±0.00 | 70.86±0.00 | 73.15±0.00 | 60.51 |
| JiGen (Carlucci et al., 2019) | ResNet18 | 53.04±0.00 | 47.51±0.00 | 71.47±0.00 | 72.79±0.00 | 61.20 |
| RSC (Huang et al., 2020) | ResNet18 | 58.42±0.00 | 47.90±0.00 | 71.63±0.00 | 74.54±0.00 | 63.12 |
| L2A-OT (Zhou et al., 2020a) | ResNet18 | **60.60±0.00** | 50.10±0.00 | **74.80±0.00** | **77.00±0.00** | **65.60** |
| LASSO-$\tau$ | ResNet18 | 59.29±0.92 | 50.77±0.27 | 73.28±0.16 | 74.48±0.30 | _64.46_ |
| LASSO-En | ResNet18 | 58.63±0.27 | **51.48±0.96** | 72.99±0.15 | 74.46±0.30 | 64.39 |
| ERM (Vapnik, 1999) | ResNet50 | 61.30±0.70 | 52.40±0.30 | 75.80±0.10 | 76.60±0.30 | 66.50 |
| MLDG (Li et al., 2018a) | ResNet50 | 61.50±0.90 | 53.20±0.60 | 75.00±1.20 | 77.50±0.40 | 66.80 |
| RSC (Huang et al., 2020) | ResNet50 | 60.70±1.40 | 51.40±0.30 | 74.80±1.10 | 75.10±1.3 0 | 65.50 |
| GroupDRO (Sagawa et al., 2019) | ResNet50 | 60.40±0.70 | 52.70±1.00 | 75.00±0.70 | 76.00±0.70 | 66.00 |
| MTL(Blanchard et al., 2021) | ResNet50 | 61.50±0.70 | 52.40±0.60 | 74.90±0.40 | 76.80±0.40 | 66.40 |
| SagNet (Nam et al., 2021) | ResNet50 | 63.40±0.20 | 54.80±0.40 | 75.80±0.40 | 78.30±0.30 | 68.10 |
| LASSO-$\tau$ | ResNet50 | 66.56±0.04 | **58.21±0.19** | **78.69±0.01** | **79.27±0.42** | **70.70** |
| LASSO-En | ResNet50 | **67.07±0.38** | 58.13±0.31 | 78.27±0.02 | 79.19±0.41 | _70.66_ |

Table 3: Classification Accuracy on PACS.

| Method | Backbone | Photo | Art-painting | Cartoon | Sketch | Average |
|---|---|---|---|---|---|---|
| ERM (Vapnik, 1999) | AlexNet | 88.47±0.63 | 67.21±0.72 | 66.12±0.51 | 55.32±0.44 | 69.28 |
| MLDG (Li et al., 2018a) | AlexNet | 88.00±0.00 | 66.23±0.00 | 66.88±0.00 | 58.96±0.00 | 70.01 |
| MASF (Dou et al., 2019) | AlexNet | 90.68±0.12 | **70.35±0.33** | **72.46±0.19** | 67.33±0.12 | **75.21** |
| JiGen (Carlucci et al., 2019) | AlexNet | 89.00±0.00 | 67.63±0.00 | _71.71±0.00_ | 65.18±0.00 | 73.38 |
| MetaReg (Balaji et al., 2018) | AlexNet | **91.07±0.41** | _69.82±0.76_ | 70.35±0.63 | 59.26±0.31 | 72.62 |
| DMG (Chattopadhyay et al., 2020) | AlexNet | 87.31±0.00 | 64.65±0.00 | 69.88±0.00 | **71.42±0.00** | 73.32 |
| LASSO-$\tau$ | AlexNet | 89.45±0.03 | 69.34±0.64 | 70.91±0.25 | 69.01±0.54 | _74.68_ |
| LASSO-En | AlexNet | 89.12±0.06 | 69.19±0.48 | 70.49±0.21 | 68.92±0.67 | 74.43 |
| ERM (Vapnik, 1999) | ResNet18 | 95.84±0.27 | 77.86±1.29 | 76.91±0.64 | 75.43±0.37 | 81.51 |
| MLDG (Li et al., 2018a) | ResNet18 | 94.30±0.00 | 79.50±0.00 | 77.30±0.00 | 71.50±0.00 | 80.70 |
| MASF (Dou et al., 2019) | ResNet18 | 94.99±0.09 | 80.29±0.18 | 77.17±0.08 | 71.69±0.22 | 81.04 |
| JiGen (Carlucci et al., 2019) | ResNet18 | _96.03±0.00_ | 79.42±0.00 | 75.25±0.00 | 71.35±0.00 | 79.14 |
| MetaReg (Balaji et al., 2018) | ResNet18 | 95.50±0.24 | **83.70±0.19** | 77.20±0.31 | 70.30±0.28 | 81.70 |
| DMG (Chattopadhyay et al., 2020) | ResNet18 | 93.35±0.00 | 76.90±0.00 | **80.38±0.00** | 75.21±0.00 | 81.46 |
| L2A-OT (Zhou et al., 2020a) | ResNet18 | **96.20±0.00** | 83.30±0.00 | 78.20±0.00 | 73.60±0.00 | 82.80 |
| RSC (Huang et al., 2020) | ResNet18 | 93.52±0.00 | 78.40±0.00 | 78.80±0.00 | **79.78±0.00** | 82.63 |
| SFA-A (Li et al., 2021) | ResNet18 | 93.90±0.00 | 81.20±0.00 | 77.80±0.00 | 73.70±0.00 | 81.70 |
| MatchDG (Mahajan et al., 2021) | ResNet18 | 95.93±0.00 | 79.77±0.00 | _80.03±0.00_ | 77.11±0.00 | **83.21** |
| LASSO-$\tau$ | ResNet18 | 94.76±0.03 | 82.17±0.60 | 78.37±0.55 | _77.34±0.37_ | _83.16_ |
| LASSO-En | ResNet18 | 94.76±0.03 | 82.02±0.75 | 77.65±0.76 | 77.07±0.42 | 82.87 |
| ERM (Vapnik, 1999) | ResNet50 | 97.2±0.30 | 84.7±0.40 | 80.8±0.60 | 79.3±1.00 | 85.50 |
| MLDG (Li et al., 2018a) | ResNet50 | 97.4±0.30 | 85.5±1.40 | 80.1±1.70 | 76.6±1.10 | 84.90 |
| MetaReg (Balaji et al., 2018) | ResNet50 | _97.6±0.31_ | 87.2±0.13 | 79.2±0.27 | 70.3±0.18 | 83.60 |
| DMG (Chattopadhyay et al., 2020) | ResNet50 | 95.0±0.00 | 82.9±0.00 | 80.5±0.00 | 72.3±0.00 | 82.67 |
| RSC (Huang et al., 2020) | ResNet50 | _97.6±0.30_ | 85.4±0.80 | 79.7±1.80 | 78.2±1.20 | 85.20 |
| GroupDRO (Sagawa et al., 2019) | ResNet50 | 96.7±0.30 | 83.5±0.90 | 79.1±0.60 | 78.3±2.00 | 84.40 |
| MTL (Blanchard et al., 2021) | ResNet50 | 96.4±0.80 | _87.5±0.80_ | 77.1±0.50 | 77.3±1.80 | 84.60 |
| SagNet (Nam et al., 2021) | ResNet50 | 97.1±0.10 | 87.4±1.00 | 80.7±0.60 | 80.0±0.40 | 86.30 |
| MatchDG (Mahajan et al., 2021) | ResNet50 | **97.94±0.00** | 85.61±0.0 | 82.12±0.00 | 78.76±0.00 | 86.11 |
| LASSO-$\tau$ | ResNet50 | 96.62±0.56 | **87.81±0.69** | **82.99±0.71** | **82.33±0.20** | **87.43** |
| LASSO-En | ResNet50 | 96.93±0.65 | 87.23±0.57 | 82.55±1.09 | 81.95±0.37 | _87.15_ |

## 4.2 ABLATION STUDY

**Sub-space representation:** To better understand the benefits of learning latent sub-spaces, we visualize the distribution of the learned features to analyze the latent space generated by ERM and LASSO-$\tau$ using t-SNE (van der Maaten & Hinton, 2008) in Figure 2. Furthermore, we decide to choose "sketch" which contains only colorless images as the target domain. This domain can be considered as the most distant domain from the others and hence results in the largest source-target divergence. Figure 2a shows feature distributions of different domains on latent space, in which samples are colored accordingly to domain label, with source domains are in red, green and back, while target domain is in blue. As it can be clearly seen, ERM training results in representations with high cross-domain separation for both target and source domains. On the other hands, the domain representations on a sub-latent space of LASSO is more homogeneously mixed, since samples from different domains are hardly distinguished. Figure 2b further shows that features from differ-

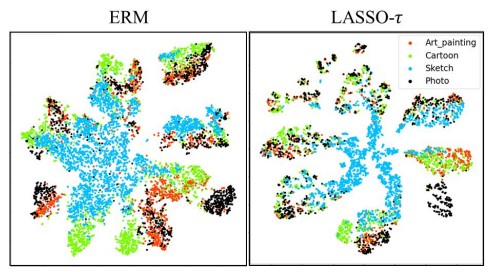
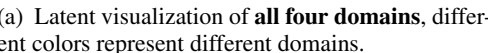
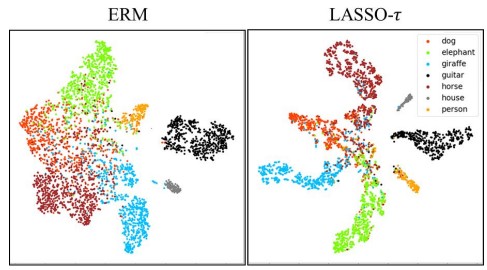

(a) Latent visualization of **all four domains**, different colors represent different domains.

(b) Latent visualization of **target domain only**, different colors represent different classes.

Figure 2: The t-SNE feature visualization of two methods ERM and LASSO-$\tau$ for experiment on PACS dataset with "Sketch" (represented by blue in 2a) as target domain using ResNet50.

ent classes are more compact and clearly separated on LASSO's sub-latent space than on full single latent space. These results indicate that feature distributions are better aligned on latent sub-space with our proposed LASSO approach.

Additionally, we further verify the data shift reduction on sub-spaces by measuring the divergence between the source and target distributions of ERM and LASSO-$\tau$ via Jensen-Shannon (JS) divergence ($\mathcal{R}_{JS}$). We employ a domain discriminator $h_d$ which distinguishes the source domain:

$$\mathcal{L}(h_d) = \mathbb{E}_{x \sim \mathbb{P}_\pi^S} \left[ \log(h_d(G(x))) \right] + \mathbb{E}_{x \sim \mathbb{P}^T} \left[ \log(1 - h_d(G(x))) \right], \qquad (9)$$

where $G$ is the feature extractor, which is $g$ for ERM and $g^\Gamma$ for LASSO-$\tau$. It is well-known that if we search $h_d$ in a family with infinite capacity then $\mathcal{R}_{JS}(G_\# \mathbb{P}_\pi^S || G_\# \mathbb{P}^T) = \max_{h^d} \frac{\mathcal{L}(h_d) + 2 \log 2}{2}$ (Goodfellow et al., 2014). The empirical results $\mathcal{R}_{JS}^{\text{ERM}} = 0.247$ and $\mathcal{R}_{JS}^{\text{LASSO-}\tau} = 0.164$ show that the latent distribution divergence between sources and target domains is decreased on sub-spaces. This observation once again provides supporting evidences for our approach.

**Effect of $\tau$:** we present the performance on validation set (Val) of source domains and target domain with different thresholds $\tau$ to analyze its effect along with the average number of selected features. The Table 4 demonstrates that by choosing reasonable $\tau$, the model performance can be improved since latent is mapped to appropriate sub-space (i.e., sub-space being compact to reduce the data shift while preserving sufficient informative features for label prediction). Alternatively, in the case of large $\tau$, the performance can be dropped since very few features remain. It also can be seen that with $\tau = 0.6$ and $\tau = 0.7$, the model utilizes

Table 4: Average number of selected features (SF) based on $\tau$ and its performance (a trial on PACS dataset with "sketch" as target domain using ResNet50).

| $\tau$ | val-SF | Val | target-SF | Target |
|---|---|---|---|---|
| 0→0.5 | 2048 | 97.24 | 2048 | 82.86 |
| 0.6 | 64.8 | 97.63 | 59.9 | 83.38 |
| 0.7 | 20.9 | 96.75 | 15.4 | 83.63 |
| 0.8 | 7.8 | 84.23 | 3.4 | 79.30 |
| 0.9 | 2.7 | 61.52 | 1.4 | 45.58 |

only a small number of features (i.e., 59.9 and 14.4 on target domain respectively) but nonetheless still yields good results. This indicates that *sub-indicator* $\Gamma$ is able to identify essential label-informative features.

## 5 CONCLUSION

In this paper, we first theoretically analyze the upper bound of target loss on latent space in domain generalization settings and the main factors that affect the performance. Then we point out that learning on appropriate latent sup-space can derive better generalization performance for domain generalization task. Motivated by this, we have proposed a novel LASSO framework which can: (1) learn diverse latent sub-spaces and corresponding individual hypotheses and (2) project source and target examples onto appropriate sub-spaces preserving crucial label-informative features for label prediction, while reducing the latent data shift due to the sub-spaces compactness compared to the entire latent space. Experiments on benchmark datasets verify that LASSO achieves competitive and even state-of-the-art performances.

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

## A  THEORETICAL DEVELOPMENT

In this Section, we present all proofs relevant to theory developed in our paper. We begin with a crucial proposition for our theory development.

**Proposition A.1.** Let $f : \mathcal{X} \mapsto \mathcal{Y}_\Delta$ where $f = h \circ g$ with $g : \mathcal{X} \mapsto \mathcal{Z}$ and $h : \mathcal{Z} \mapsto \mathcal{Y}_\Delta$.

i) We have

$$\sum_{y=1}^{C} \int \ell\left(h\left(z\right), y\right) p^g\left(y \mid z\right) p^g\left(z\right) dz = \sum_{y=1}^{C} \int \ell\left(f\left(x\right), y\right) p\left(y \mid x\right) p\left(x\right) dx,$$

where $p^g\left(y \mid z\right) = \frac{\int_{g^{-1}(z)} p(y|x)p(x)dx}{\int_{g^{-1}(z)} p(x)dx}$ for any $z \in g\left(\mathcal{X}\right)$ and $p^g\left(z\right) = \int_{g^{-1}(z)} p\left(x\right) dx$ is the density of the push-forward distribution $\mathbb{P}^g = g_\#\mathbb{P}$.

ii) We have

$$\sum_{y=1}^{C} \int \left|p^{1,g}\left(y \mid z\right) - p^{2,g}\left(y \mid z\right)\right| p^g\left(z\right) dz \leq \sum_{y=1}^{C} \int \left|p^1\left(y \mid x\right) - p^2\left(y \mid x\right)\right| p\left(x\right) dx,$$

where $p^g\left(z\right) = \int_{g^{-1}(z)} p\left(x\right) dx$ is the density of the push-forward distribution $\mathbb{P}^g = g_\#\mathbb{P}$ and $p^{1,g}\left(y \mid z\right) = \frac{\int_{g^{-1}(z)} p^1(y|x)p(x)dx}{\int_{g^{-1}(z)} p(x)dx}$ and $p^{2,g}\left(y \mid z\right) = \frac{\int_{g^{-1}(z)} p^2(y|x)p(x)dx}{\int_{g^{-1}(z)} p(x)dx}$ for any $z \in g\left(\mathcal{X}\right)$.

*Proof.* i) We first prove for any $y \in [C]$ that

$$\int \ell\left(h\left(z\right), y\right) p^g\left(y \mid z\right) p^g\left(z\right) dz = \int \ell\left(f\left(x\right), y\right) p\left(y \mid x\right) p\left(x\right) dx.$$

In fact, we have

$$
\begin{aligned}
\int \ell\left(h\left(z\right),y\right)p^g\left(y\mid z\right)p^g\left(z\right)dz &= \int_{\mathcal{Z}} \ell\left(h\left(z\right),y\right)\frac{\int_{g^{-1}(z)} p\left(y\mid x\right)p\left(x\right)dx}{\int_{g^{-1}(z)} p\left(x\right)dx}\int_{g^{-1}(z)} p\left(x\right)dxdz \\
&= \int_{\mathcal{Z}} \ell\left(h\left(z\right),y\right)\int_{g^{-1}(z)} p\left(y\mid x\right)p\left(x\right)dxdz \\
&= \int_{\mathcal{Z}}\int_{g^{-1}(z)} \ell\left(h\left(z\right),y\right)p\left(y\mid x\right)p\left(x\right)dxdz \\
&= \int_{\mathcal{Z}}\int_{\mathcal{X}} \mathbf{1}_{x\in g^{-1}(z)}\ell\left(h\left(z\right),y\right)p\left(y\mid x\right)p\left(x\right)dxdz \\
&\overset{(1)}{=} \int_{\mathcal{X}}\int_{\mathcal{Z}} \mathbf{1}_{x\in g^{-1}(z)}\ell\left(h\left(z\right),y\right)p\left(y\mid x\right)p\left(x\right)dzdx \\
&= \int_{\mathcal{X}}\int_{\mathcal{Z}} \mathbf{1}_{z=g(x)}\ell\left(h\left(z\right),y\right)p\left(y\mid x\right)p\left(x\right)dzdx \\
&= \int_{\mathcal{X}} \ell\left(h\left(g\left(x\right)\right),y\right)p\left(y\mid x\right)p\left(x\right)dx \\
&= \int \ell\left(f\left(x\right),y\right)p\left(y\mid x\right)p\left(x\right)dx.
\end{aligned}
$$

where note that $\mathbf{1}_{\text{condition}}$ is the indicator function which returns 1 if condition is true and 0 otherwise, the equality in $\overset{(1)}{=}$ due to Fubini theorem.

Finally, we reach

$$
\sum_{y=1}^{C}\int \ell\left(h\left(z\right),y\right)p^g\left(y\mid z\right)p^g\left(z\right)dz = \sum_{y=1}^{C}\int \ell\left(f\left(x\right),y\right)p\left(y\mid x\right)p\left(x\right)dx.
$$

ii) We then prove that

$$
\int \left|p^{1,g}\left(y\mid z\right)-p^{2,g}\left(y\mid z\right)\right|p^g\left(z\right)dz \le \int \left|p^{1}\left(y\mid x\right)-p^{2}\left(y\mid x\right)\right|p\left(x\right)dx, \forall y\in[C].
$$

We derive as

$$
\begin{aligned}
&\int_{\mathcal{Z}} \left|p^{1,g}\left(y\mid z\right)-p^{2,g}\left(y\mid z\right)\right|p^g\left(z\right)dz \\
&= \int_{\mathcal{Z}} \left|\frac{\int_{g^{-1}(z)} p^{1}\left(y\mid x\right)p\left(x\right)dx}{\int_{g^{-1}(z)} p\left(x\right)dx}-\frac{\int_{g^{-1}(z)} p^{2}\left(y\mid x\right)p\left(x\right)dx}{\int_{g^{-1}(z)} p\left(x\right)dx}\right|\int_{g^{-1}(z)} p\left(x\right)dxdz \\
&= \int_{\mathcal{Z}} \left|\int_{g^{-1}(z)} p^{1}\left(y\mid x\right)p\left(x\right)dx-\int_{g^{-1}(z)} p^{2}\left(y\mid x\right)p\left(x\right)dx\right|dz \\
&= \int_{\mathcal{Z}} \left|\int_{g^{-1}(z)} \left[p^{1}\left(y\mid x\right)-p^{2}\left(y\mid x\right)\right]p\left(x\right)dx\right|dz \\
&\le \int_{\mathcal{Z}}\int_{g^{-1}(z)} \left|p^{1}\left(y\mid x\right)-p^{2}\left(y\mid x\right)\right|p\left(x\right)dxdz \\
&= \int_{\mathcal{Z}}\int_{\mathcal{X}} \mathbf{1}_{x\in g^{-1}(z)}\left|p^{1}\left(y\mid x\right)-p^{2}\left(y\mid x\right)\right|p\left(x\right)dxdz \\
&\overset{(1)}{=} \int_{\mathcal{X}}\int_{\mathcal{Z}} \mathbf{1}_{z=g(x)}\left|p^{1}\left(y\mid x\right)-p^{2}\left(y\mid x\right)\right|p\left(x\right)dzdx \\
&= \int_{\mathcal{X}} \left|p^{1}\left(y\mid x\right)-p^{2}\left(y\mid x\right)\right|p\left(x\right)dx.
\end{aligned}
$$

where in the derivation of $\overset{(1)}{=}$, we use Fubini theorem again. $\qquad\square$

.

**Theorem 4.** *(**Theorem 1 in the main paper**) If the loss function $\ell$ is upper-bounded by a positive constant L, the target general loss is upper-bounded by:*

$$\mathcal{L}(f, \mathbb{D}^T) \leq \exp\left\{R^\alpha\left(g_\# \mathbb{P}^T, g_\# \mathbb{P}^S_\pi\right)\right\}^{\frac{\alpha-1}{\alpha}} L^{\frac{1}{\alpha}}\left[\mathcal{L}\left(f, \mathbb{D}^S_\pi\right) + L \max_k \mathbb{E}_{\mathbb{P}^S_k}\left[\|\Delta p_k\left(y \mid \boldsymbol{x}\right)\|_1\right]\right]^{\frac{\alpha-1}{\alpha}},$$

(10)

*where $g_\# \mathbb{P}^S_\pi$ and $g_\# \mathbb{P}^T$ are pushed-forward distributions induced by applying g on $\mathbb{P}^S_\pi$ and $\mathbb{P}^T$, $R^\alpha$ is the $\alpha$-divergence ($\alpha > 1$), and $\Delta p_k\left(y \mid \boldsymbol{x}\right) := \left[\left|p^S_k\left(y = c \mid x\right) - p^T\left(y = c \mid x\right)\right|\right]^C_{c=1}$ represents the label shift between the labeling assignment mechanisms of an individual source domain and target domain on the input space.*

*Proof.* Let $\mathbb{D}^{S,g}_\pi$ be the joint distribution including the pairs $(y, z)$ where $k \sim Cat\left(\pi\right)$, $z \sim p^{S,g}_k\left(z\right)$, and $y \sim p^{S,g}_k\left(y \mid z\right)$. The density $p^{S,g}_\pi\left(y, z\right)$ of this distribution is as follows:

$$p^{S,g}_\pi\left(y, z\right) = \sum_{k=1}^K \pi_k p^{S,g}_k\left(z\right) p^{S,g}_k\left(y \mid z\right) = \sum_{k=1}^K \pi_k p^{S,g}_k\left(y, z\right).$$

Let $\mathbb{D}^{T,g}$ be the joint distribution including the pairs $(y, z)$ where $z \sim p^{T,g}\left(z\right)$ and $y \sim p^{T,g}\left(y \mid z\right)$. The density $p^{T,g}(y, z)$ of this distribution is as follows:

$$p^{T,g}(y, z) = p^{T,g}\left(z\right) p^{T,g}\left(y \mid z\right).$$

Eventually, we denote $\mathbb{D}^{h,g}$ as the joint distribution including the pairs $(y, z)$ where $z \sim p^{S,g}_\pi\left(z\right) := \sum_{k=1}^K \pi_k p^{S,g}_k\left(z\right)$ and $y \sim p^{T,g}\left(y \mid z\right)$. The density $p^{h,g}\left(y, z\right)$ of this distribution is as follows:

$$p^{h,g}\left(y, z\right) = p^{S,g}_\pi\left(z\right) p^{T,g}\left(y \mid z\right) = \sum_{k=1}^K \pi_k p^{S,g}_k\left(z\right) p^{T,g}\left(y \mid z\right).$$

With the above equipment, we have

$$\mathcal{L}\left(h, \mathbb{D}^{h,g}\right) = \int \ell\left(h\left(z\right), y\right) p^{h,g}\left(z, y\right) dz dy$$

$$= \mathcal{L}\left(h, \mathbb{D}^{S,g}_\pi\right) + \int \ell\left(h\left(z\right), y\right)\left[p^{h,g}\left(z, y\right) - p^{S,g}_\pi\left(z, y\right)\right] dz dy$$

$$\leq \mathcal{L}\left(h, \mathbb{D}^{S,g}_\pi\right) + \int \ell\left(h\left(z\right), y\right)|p^{h,g}\left(z, y\right) - p^{S,g}_\pi\left(z, y\right)|dz dy$$

$$\leq \mathcal{L}\left(h, \mathbb{D}^{S,g}_\pi\right) + L\int|p^{h,g}\left(z, y\right) - p^{S,g}_\pi\left(z, y\right)|dz dy$$

$$= \mathcal{L}\left(h, \mathbb{D}^{S,g}_\pi\right) + L\int \sum_{k=1}^K \pi_k p^{S,g}_k\left(z\right)|p^{T,g}\left(y \mid z\right) - p^{S,g}_k\left(y \mid z\right)|dz dy$$

$$= \mathcal{L}\left(h, \mathbb{D}^{S,g}_\pi\right) + L\sum_{y=1}^C \sum_{k=1}^K \pi_k \int|p^{T,g}\left(y \mid z\right) - p^{S,g}_k\left(y \mid z\right)|p^{S,g}_k\left(z\right) dz.$$

We further note that

$$
\begin{aligned}
\mathcal{L}\left(h, \mathbb{D}_\pi^{S,g}\right) &= \sum_{y=1}^{C} \int \ell\left(h(z), y\right) p_\pi^{S,g}\left(y, z\right) dz \\
&= \sum_{y=1}^{C} \sum_{k=1}^{K} \pi_k \int \ell\left(h(z), y\right) p_k^{S,g}\left(y \mid z\right) p_k\left(z\right) dz \\
&= \sum_{k=1}^{K} \pi_k \sum_{y=1}^{C} \int \ell\left(h(z), y\right) p_k^{S,g}\left(y \mid z\right) p_k\left(z\right) dz \\
&\overset{(1)}{=} \sum_{k=1}^{K} \sum_{y=1}^{C} \pi_k \int \ell\left(f(x), y\right) p_k^{S}\left(y \mid x\right) p_k\left(x\right) dx = \mathcal{L}\left(f, \mathbb{D}_\pi^S\right),
\end{aligned}
$$

where we have $\overset{(1)}{=}$ by using Proposition A.1 (i).

$$
\begin{aligned}
&\sum_{y=1}^{C} \sum_{k=1}^{K} \pi_k \int \left|p^{T,g}\left(y \mid z\right) - p_k^{S,g}\left(y \mid z\right)\right| p_k^{S,g}\left(z\right) dz \\
&= \sum_{k=1}^{K} \pi_k \sum_{y=1}^{C} \int \left|p^{T,g}\left(y \mid z\right) - p_k^{S,g}\left(y \mid z\right)\right| p_k^{S,g}\left(z\right) dz \\
&\overset{(2)}{\leq} \sum_{k=1}^{K} \pi_k \sum_{y=1}^{C} \int \left|p^{T}\left(y \mid x\right) - p_k^{S}\left(y \mid x\right)\right| p_k^{S}\left(x\right) dx \\
&= \sum_{k=1}^{K} \pi_k \int \sum_{y=1}^{C} \left|p^{T}\left(y \mid x\right) - p_k^{S}\left(y \mid x\right)\right| p_k^{S}\left(x\right) dx \\
&= \sum_{k=1}^{K} \pi_k \mathbb{E}_{\mathbb{P}_k^S}\left[\left\|\Delta p_k\left(y \mid \mathbf{x}\right)\right\|_1\right] \leq \max_k \mathbb{E}_{\mathbb{P}_k^S}\left[\left\|\Delta p_k\left(y \mid \mathbf{x}\right)\right\|_1\right],
\end{aligned}
$$

where we have $\overset{(2)}{=}$ by using Proposition A.1 (ii).

Therefore, we obtain

$$
\mathcal{L}\left(h, \mathbb{D}^{h,g}\right) \leq \mathcal{L}\left(f, \mathbb{D}_\pi^S\right) + \max_k \mathbb{E}_{\mathbb{P}_k^S}\left[\left\|\Delta p_k\left(y \mid \mathbf{x}\right)\right\|_1\right]. \tag{11}
$$

Finally, we manipulate $\mathcal{L}\left(h, \mathbb{D}^{T,g}\right)$ as

$$
\begin{aligned}
\mathcal{L}\left(h, \mathbb{D}^{T,g}\right) &= \int_{\mathcal{Y} \times \mathcal{Z}} \ell\left(h\left(z\right), y\right) p^{T,g}\left(z, y\right) dz dy \\
&= \int_{\mathcal{Y} \times \mathcal{Z}} \frac{p^{T,g}\left(z, y\right)}{p^{h,g}\left(z, y\right)^{\frac{\alpha-1}{\alpha}}} p^{h,g}\left(z, y\right)^{\frac{\alpha-1}{\alpha}} \ell\left(h\left(z\right), y\right) dz dy.
\end{aligned}
$$

The Holder inequality gives us

$$
\mathcal{L}\left(h, \mathbb{D}^{T,g}\right) \leq \left[\int_{\mathcal{Y} \times \mathcal{Z}} \frac{p^{T,g}\left(z, y\right)^{\alpha}}{p^{h,g}\left(z, y\right)^{\alpha-1}} dz dy\right]^{\frac{1}{\alpha}} \left[\int_{\mathcal{Y} \times \mathcal{Z}} p^{h,g}\left(z, y\right) \ell\left(h\left(z\right), y\right)^{\frac{\alpha}{\alpha-1}} dz dy\right]^{\frac{\alpha-1}{\alpha}}.
$$

Referring to the definition of the Rényi divergence and note that $\ell\left(h\left(z\right), y\right) \leq L$, we obtain

$$
\mathcal{L}\left(h, \mathbb{D}^{T,g}\right) \leq \left[\exp\left\{R^\alpha\left(\mathbb{D}^{T,g} \| \mathbb{D}^{h,g}\right)\right\} \mathcal{L}\left(h, \mathbb{D}^{h,g}\right)\right]^{\frac{\alpha-1}{\alpha}} L^{\frac{1}{\alpha}}. \tag{12}
$$

We further derive

$$
\begin{aligned}
R^{\alpha}\left(\mathbb{D}^{T,g}\|\mathbb{D}^{h,g}\right) &= \frac{1}{\alpha-1}\log\left(\int\left[\frac{p^{T,g}\left(z,y\right)}{p^{h,g}\left(z,y\right)}\right]^{\alpha-1}p^{T,g}\left(z,y\right)dzdy\right) \\
&= \frac{1}{\alpha-1}\log\left(\int\left[\frac{p^{T,g}\left(z\right)p^{T,g}\left(y\mid z\right)}{p_{\pi}^{S,g}\left(z\right)p^{T,g}\left(y\mid z\right)}\right]^{\alpha-1}p^{T,g}\left(z,y\right)dzdy\right) \\
&= \frac{1}{\alpha-1}\log\left(\int\left[\frac{p^{T,g}\left(z\right)}{p_{\pi}^{S,g}\left(z\right)}\right]^{\alpha-1}p^{T,g}\left(z,y\right)dzdy\right) \\
&= \frac{1}{\alpha-1}\log\left(\int\left[\frac{p^{T,g}\left(z\right)}{p_{\pi}^{S,g}\left(z\right)}\right]^{\alpha-1}\sum_{y=1}^{C}p^{T,g}\left(z,y\right)dz\right) \\
&= \frac{1}{\alpha-1}\log\left(\int\left[\frac{p^{T,g}\left(z\right)}{p_{\pi}^{S,g}\left(z\right)}\right]^{\alpha-1}p^{T,g}\left(z\right)dz\right) \\
&= R^{\alpha}\left(g_{\#}\mathbb{P}^{T}\|g_{\#}\mathbb{P}_{\pi}^{S}\right).
\end{aligned}
\tag{13}
$$

Therefore, combining (12) and (13), we reach the following inequality:

$$
\mathcal{L}\left(h,\mathbb{D}^{T,g}\right)\leq\left[\exp\left\{R^{\alpha}\left(g_{\#}\mathbb{P}^{T}\|g_{\#}\mathbb{P}_{\pi}^{S}\right)\right\}\mathcal{L}\left(h,\mathbb{D}^{h,g}\right)\right]^{\frac{\alpha-1}{\alpha}}L^{\frac{1}{\alpha}}.
$$

Subsequently, by noting that

$$
\begin{aligned}
\mathcal{L}\left(h,\mathbb{D}_{\pi}^{T,g}\right) &= \sum_{y=1}^{C}\int\ell\left(h(z),y\right)p^{T,g}\left(y\mid z\right)p^{T,g}\left(z\right)dz \\
&\overset{(1)}{=}\sum_{y=1}^{C}\int\ell\left(f\left(x\right),y\right)p^{T}\left(y\mid x\right)p^{T}\left(x\right)dx=\mathcal{L}\left(f,\mathbb{D}^{T}\right),
\end{aligned}
$$

we reach

$$
\mathcal{L}\left(f,\mathbb{D}^{T}\right)\leq\left[\exp\left\{R^{\alpha}\left(g_{\#}\mathbb{P}^{T}\|g_{\#}\mathbb{P}_{\pi}^{S}\right)\right\}\mathcal{L}\left(h,\mathbb{D}^{h,g}\right)\right]^{\frac{\alpha-1}{\alpha}}L^{\frac{1}{\alpha}}.
$$

Finally, referring to the inequality in (11), we reach the conclusion. $\qquad\square$

**Remark:** We would like to highlight that our Theorem 1 despite developing in a general context of DG is also novel due to two following reasons: (i) it proposes the target loss upper-bound in a general setting of multi-class classification with a sufficiently general loss, which can be viewed as a non-trivial generalization of existing works in DA and DG (Mansour et al., 2009; Ben-David et al., 2010; Zhao et al., 2019) and (ii) our bound is interweaving both input and latent spaces, which is novel and appropriate for theoretical analyses in deep learning, while the bounds in previous works only involve input space.

**Theorem 5.** *(Theorem 2 in the main paper) Given a deterministic sub-space indicator $\Gamma$, if the loss function $\ell$ is upper-bounded by a positive constant $L$, the sub-space target general loss is upper-bounded by:*

$$
\mathcal{L}\left(f_{\mathcal{M}},\mathbb{D}^{T}\right)\leq\frac{L^{\frac{1}{\alpha}}}{|\mathcal{M}|}\sum_{m\in\mathcal{M}}\exp\left\{R^{\alpha}\left(g^{\Gamma}{}_{\#}\mathbb{P}_{m}^{T},g^{\Gamma}{}_{\#}\mathbb{P}_{m}^{S}\right)\right\}^{\frac{\alpha-1}{\alpha}}\left[\mathcal{L}\left(f_{m},\mathbb{D}_{m}^{S}\right)+L\Delta p_{m}\right]^{\frac{\alpha-1}{\alpha}},\tag{14}
$$

*where $\mathbb{D}_{m}^{S}$ is the joint distribution of $(x,y)$ with $x\sim\mathbb{P}_{m}^{S}$ and $y\sim p_{m}^{S}\left(\cdot\mid x\right)$, and $\Delta p_{m}:=\mathbb{E}_{x\sim\mathbb{P}_{m}^{S}}\left[\left\|p_{m}^{S}\left(\cdot\mid x\right)-p_{m}^{T}\left(\cdot\mid x\right)\right\|_{1}\right]$. sub-space indices $m$.*

*Proof.* Given a sub-space index $m\in\mathcal{M}$, by noting that $f_{m}\left(x\right)=h\left(g^{\Gamma}\left(x\right)\right)$ with $x\sim\mathbb{P}_{m}^{S}$ over $A_{m}$, using the same proof for a single space in Theorem 4, we obtain

$$
\mathcal{L}\left(f_{m},\mathbb{D}^{T}\right)\leq L^{\frac{1}{\alpha}}\exp\left\{R^{\alpha}\left(g^{\Gamma}{}_{\#}\mathbb{P}_{m}^{T},g^{\Gamma}{}_{\#}\mathbb{P}_{m}^{S}\right)\right\}^{\frac{\alpha-1}{\alpha}}\left[\mathcal{L}\left(f_{m},\mathbb{D}_{m}^{S}\right)+L\Delta p_{m}\right]^{\frac{\alpha-1}{\alpha}}.
$$

Finally, taking average over $m \in \mathcal{M}$, we reach

$$\mathcal{L}\left(f_{\mathcal{M}}, \mathbb{D}^T\right) \leq \frac{L^{\frac{1}{\alpha}}}{|\mathcal{M}|} \sum_{m \in \mathcal{M}} \exp\left\{R^\alpha\left(g^\Gamma{}_\# \mathbb{P}_m^T, g^\Gamma{}_\# \mathbb{P}_m^S\right)\right\}^{\frac{\alpha-1}{\alpha}} \left[\mathcal{L}\left(f_m, \mathbb{D}_m^S\right) + L\Delta p_m\right]^{\frac{\alpha-1}{\alpha}}.$$

$\square$

**Theorem 6.** *(**Theorem 3 in the main paper**) Let $X$ is a random variable of source sample (i.e., drawn from $\mathbb{P}_\pi^S$) and $Y$ is a random variable of ground-truth labels. Denote $N = \sum_{m' \in \mathcal{M}} \mathbb{P}_\pi^S\left(A_{m'}\right)$, we then have*

$$\mathbb{I}\left(\Gamma\left(g\left(X\right)\right) \odot g\left(X\right), Y\right) \geq -\sum_{m \in \mathcal{M}} \frac{\mathbb{P}_\pi^S\left(A_m\right)}{N} \mathcal{L}\left(f_m, \mathbb{D}_m^S\right) + const, \tag{15}$$

*where the loss $\mathcal{L}\left(f_m, \mathbb{D}_m^S\right)$ is defined based on the cross-entropy loss and $\mathbb{I}$ denotes the mutual information.*

*Proof.* Denote $T = \Gamma\left(g\left(X\right)\right) \odot g\left(X\right)$, we have

$$\begin{aligned}
\mathbb{I}\left(T, Y\right) &= \int p(t, y) \log \frac{p\left(t, y\right)}{p\left(t\right) p\left(y\right)} dt dy \\
&= \int p(t, y) \log \frac{p\left(y \mid t\right)}{p\left(y\right)} dt dy \\
&= \int p(t, y) \log p\left(y \mid t\right) dt dy + \mathbb{H}\left(Y\right) \\
&= \int p(t, y) \log h\left(y \mid t\right) \frac{p\left(y \mid t\right)}{h\left(y \mid t\right)} dt dy + \mathbb{H}\left(Y\right) \\
&= \int p(t, y) \log h\left(t, y\right) dt dy + D_{KL}\left(p\left(y \mid t\right) \| h\left(y \mid t\right)\right) + \mathbb{H}\left(Y\right) \\
&\geq \int p(t, y) \log h\left(t, y\right) dt dy + const,
\end{aligned}$$

where $\mathbb{H}$ specifies the entropy, $D_{KL}$ is Kullback-Leibler (KL) divergence, $h\left(y \mid t\right) = h\left(t, y\right) = h\left(\Gamma\left(g\left(x\right)\right) \odot g\left(x\right), y\right)$ for any $h : \mathcal{Z} \rightarrow \mathcal{Y}_\Delta$, and $h\left(t, y\right)$ returns the $y$-th element of $h\left(t\right)$.

We further derive

$$\begin{aligned}
\int p(t, y) \log h\left(t, y\right) dt dy &= \sum_{i=1}^C \mathbb{E}_{p(t)}\left[p\left(y = i \mid t\right) \log h\left(t, y = i\right)\right] \\
&\overset{(1)}{=} \sum_{i=1}^C \mathbb{E}_{p_\pi^S(x)}\left[p\left(y = i \mid \Gamma\left(g\left(x\right)\right) \odot g\left(x\right)\right) \log h\left(\Gamma\left(g\left(x\right)\right) \odot g\left(x\right), i\right)\right] \\
&= \sum_{i=1}^C \mathbb{E}_{\mathbb{P}_\pi^S}\left[p\left(y = i \mid \Gamma\left(g\left(x\right)\right) \odot g\left(x\right)\right) \log h\left(\Gamma\left(g\left(x\right)\right) \odot g\left(x\right), i\right)\right].
\end{aligned}$$

Note that we have $\overset{(1)}{=}$ because $\Gamma\left(g\left(x\right)\right) \odot g\left(x\right)$ pushes forward $X \sim p_\pi^S(x)$ to $T \sim p\left(t\right)$. Moreover, according to our definitions: $\mathbb{P}_\pi^S = \sum_{m \in \mathcal{M}} \frac{\mathbb{P}_\pi^S\left(A_m\right)}{N} \mathbb{P}_m^S$, we hence obtain

$$\begin{aligned}
\int p(t, y) \log h\left(t, y\right) dt dy &= \sum_{m \in \mathcal{M}} \frac{\mathbb{P}_\pi^S\left(A_m\right)}{N} \sum_{i=1}^C \mathbb{E}_{\mathbb{P}_m^S}\left[p\left(y = i \mid m \odot g\left(x\right)\right) \log h\left(m \odot g\left(x\right), i\right)\right] \\
&= -\sum_{m \in \mathcal{M}} \frac{\mathbb{P}_\pi^S\left(A_m\right)}{N} \sum_{i=1}^C \mathbb{E}_{\mathbb{P}_m^S}\left[-p_m^S\left(y = i \mid x\right) \log f_m\left(x, i\right)\right] \\
&= -\sum_{m \in \mathcal{M}} \frac{\mathbb{P}_\pi^S\left(A_m\right)}{N} \mathcal{L}\left(f_m, \mathbb{D}_m^S\right).
\end{aligned}$$

$\square$

## B  ADDITIONAL EXPERIMENTS

### B.1  EFFECT OF THE ATTRIBUTE-BASED INDICATOR $\Gamma_d$

We consider a feature vector $z = [z_1, ..., z_D] \in \mathbb{R}^{D' \times D}$, whereas each $z_d \in \mathbb{R}^{D'}$ known as *an attribute* and model sub-space indicator as $\Gamma(z) = [\Gamma_1(z_1), ..., \Gamma_D(z_D)] \in [0, 1]^D$, $\Gamma_d(z) = \Gamma_d(z_d)$ is computed based solely on the group of $D'$ features in the attribute $z_d$ itself rather than full latent $z$. Consequently, attribute-based $\Gamma_d$ depends on attributes which are shared across domains, hence, becomes more independent to domain. Moreover, by Theorem 3, model can learn meaningful attributes as long as the performance on source domains is guaranteed. In unseen domains, the target of sub-space indicator $\Gamma$ is to detect label-informative attributes which is learned in source domains instead of identifying all label-informative attributes.

More specifically, in training phase, since attributes $z_d$ come from different source domains, corresponding $\Gamma_d$ plays as a domain invariant hypothesis which output $\{0, 1\}$ identifying label-informative attributes across source domains. In unseen domain, $\Gamma_d(z_d)$ only predict $z_d$ is label-informative attribute *which are learned in source domains* or not.

Table 5: Performance LASSO-$\tau$ with different $\Gamma_d$ settings.

| Method | Backbone | Photo | Art-painting | Cartoon | Sketch | Average |
|---|---|---|---|---|---|---|
| LASSO-$\tau$ $\Gamma_d(z)$ | ResNet50 | 97.01±0.16 | 87.30±0.52 | 81.91±0.46 | 80.02±0.45 | 86.56 |
| LASSO-$\tau$ $\Gamma_d(z_d)$ | ResNet50 | 96.62±0.56 | 87.81±0.69 | 82.99±0.71 | 82.33±0.20 | 87.43 |

In order to analyze the benefit of attribute-based $\Gamma_d$, we examine the performance on the setting of $\Gamma_d(z)$ which uses full latent $z$ as input for $\Gamma_d$ and $\Gamma_d(z_d)$ which uses only attribute $z_d$ as input for $\Gamma_d$. As can be seen from Table 5, both $\Gamma_d(z_d)$ and $\Gamma_d(z)$ achieve competitive performances on target domain: "Photo", "Art-painting" and "cartoon" domain but $\Gamma_d(z_d)$ setting outperforms on "sketch" domain. Remind that using "sketch" as target domain leads to the largest source-target divergence. The results of "sketch" domain demonstrate that $\Gamma_d(z)$ becomes less efficient when source-target divergence. In contrast, $\Gamma_d(z)$ shows its better generalization as not be interfered by other noisy attributes.

### B.2  MASK SAMPLING FOR MODEL TRAINING

In training phase, sampling $m \sim Ber(\Gamma(z))$ would help hypotheses explore more useful sub-spaces. The motivation is that target domains might have or share a small sub-set of learned features in source domains e.g. in PACS dataset, many label-informative features can be learned in rich information domain such as "Photo" domain while "Sketch" domain only has edge/shape information for classification. Therefore, in training phase, we would like to train hypotheses on different sub-set of learned features (i.e., sub-spaces of sub-space) which would strengthen sub-space hypotheses on arbitrary target domains..

Table 6: Classification Accuracy on PACS with ResNet50

| Training | Backbone | Photo | Art-painting | Cartoon | Sketch | Average |
|---|---|---|---|---|---|---|
| Deterministic-$\tau$ | ResNet50 | 95.99±0.42 | 86.28±1.70 | 80.09±1.54 | 79.01±4.44 | 85.34 |
| Deterministic-En | ResNet50 | 93.05±0.03 | 86.18±1.08 | 80.08±1.62 | 77.73±2.65 | 84.26 |
| LASSO-$\tau$ | ResNet50 | 96.62±0.56 | 87.81±0.69 | 82.99±0.71 | 82.33±0.20 | **87.43** |
| LASSO-En | ResNet50 | 96.93±0.65 | 87.23±0.57 | 82.55±1.09 | 81.95±0.37 | 87.15 |

To demonstrate the benefit of sampling, we conduct additional experiments using deterministic manner which uses threshold $\tau$ to determine mask $m$ in model training. The results in Table 6 indicate sampling significantly enhance performance.

### B.3  DISCUSSION AND COMPARISON WITH RANDOM DROPOUT

We would like to report additional experimental results in comparison with "Random Dropout" Baseline (Srivastava et al., 2014) for different dropping rate as follows:

Table 7: Classification Accuracy on PACS with ResNet18

| Feature Dropping Rate | Backbone | Photo | Art-painting | Cartoon | Sketch | Average |
|---|---|---|---|---|---|---|
| 0% | ResNet18 | 96.17±0.27 | 78.42±1.29 | 76.73±0.64 | 75.40±0.37 | 81.68 |
| 10% | ResNet18 | 96.29±0.66 | 81.01±0.33 | 76.98±0.67 | 75.60±0.81 | 82.47 |
| 30% | ResNet18 | 96.41±0.60 | 79.73±0.69 | 77.68±0.28 | 75.22±1.07 | 82.26 |
| 50% | ResNet18 | 96.59±0.14 | 81.20±0.02 | 77.92±0.24 | 75.98±0.17 | 82.92 |
| 70% | ResNet18 | 96.29±0.14 | 79.93±0.72 | 78.26±0.67 | 74.89±0.09 | 82.34 |
| LASSO-$\tau$ | ResNet18 | 94.76±0.03 | 82.17±0.60 | 78.37±0.55 | 77.34±0.37 | **83.16** |
| LASSO-En | ResNet18 | 94.76±0.03 | 82.02±0.75 | 77.65±0.76 | 77.07±0.42 | 82.87 |

Table 8: Classification Accuracy on PACS with ResNet50

| Feature Dropping Rate | Backbone | Photo | Art-painting | Cartoon | Sketch | Average |
|---|---|---|---|---|---|---|
| 0% | ResNet50 | 97.20±0.30 | 84.70±0.40 | 80.80±0.60 | 79.30±1.00 | 85.50 |
| 10% | ResNet50 | 97.96±0.26 | 86.96±0.37 | 78.92±1.44 | 79.96±0.64 | 85.95 |
| 30% | ResNet50 | 98.08±0.32 | 87.26±0.63 | 80.11±1.10 | 81.31±0.80 | 86.69 |
| 50% | ResNet50 | 97.60±0.04 | 87.16±0.13 | 80.46±0.24 | 80.80±0.77 | 86.51 |
| 70% | ResNet50 | 97.08±0.47 | 86.33±0.73 | 81.78±0.98 | 80.57±1.01 | 86.44 |
| LASSO-$\tau$ | ResNet50 | 96.62±0.56 | 87.81±0.69 | 82.99±0.71 | 82.33±0.20 | **87.43** |
| LASSO-En | ResNet50 | 96.93±0.65 | 87.23±0.57 | 82.55±1.09 | 81.95±0.37 | 87.15 |

The results in Table 7 and Table 8 indicate that ERM with an appropriate dropping rate (i.e., 50% for ResNet18 and 30% for ResNet50) is able to achieve comparative performance to most current baselines.

In fact, "Dropout" has a quite similar effect to the representation as LASSO, which increases the diversity of latent representations. However, LASSO goes one step further compared to "Dropout". Specifically, representation is used as it is during inference time, without making the use of information in the target data, e.g., using the full network for inference, or average over a set of randomly "Dropout" networks (MC dropout). On the contrary, the sub-space indicator in LASSO additionally takes into account the information from the target domain to select the suitable masked network which is likely to generalize better to this particular target domain. This is the motivation behind the selection of appropriate sub-space.

## C   EXPERIMENTAL SETTINGS

### C.1   DATASET DETAILS

To evaluate the effectiveness of the proposed method, we utilize three datasets: PACS (Li et al., 2017), VLCS (Torralba & Efros, 2011), and Office-Home (Venkateswara et al., 2017), which are the common DG benchmarks with multi-source domains.

- **PACS** (Li et al., 2017): 9991 images of seven classes in total, over four domains:Art_painting (A), Cartoon (C), Sketches (S), and Photo (P).

- **VLCS** (Torralba & Efros, 2011): five classes over four domains with a total of 10729 samples. The domains are defined by four image origins, i.e., images were taken from the PASCAL VOC 2007 (V), LabelMe (L), Caltech (C) and Sun (S) datasets.

- **Office-Home** (Venkateswara et al., 2017): 65 categories of 15500 daily objects from 4 domains: Art, Clipart, Product (vendor website with white-background) and Real-World (real-object collected from regular cameras).

### C.2   EVALUATION PROTOCOL

For PACS and VLCS, we follow the setting of (Li et al., 2017) as these two datasets provide specific "train" and "validation" splits for each domain to ensure a fair comparison. We use the validation subsets from all training domains to create an overall validation set. On the target domain, we evaluate on the entire dataset. Office-Home also follows the leave-one-domain-out protocol in (Li et al., 2017). In addition, the generalization of our method is based on the accuracy of the *sub-space indicator* $\Gamma$. Therefore, we choose the model when the sub-space indicator $\Gamma$ converged. In

particular, we train all models for the same fixed number of steps and consider only the final checkpoint. Finally, we report the average performance with error bars over 5 runs for all experimental settings and compare two variants of our proposed method LASSO$-\tau$ (the threshold variant) and LASSO-En (the ensemble variant) with the recent domain generalization baselines.

## C.3 BASELINES

We compare with the following recent domain generalization baselines (with reproducible public codes):

- **ERM** (Vapnik, 1999):minimizes the sum of errors across domains and examples. For our experiments, we employ the implementation from (Gulrajani & Lopez-Paz, 2021), a strong baseline that can achieve competitive accuracies on DG benchmarks.
- **MLDG** (Li et al., 2018a): Learning to generalize: Meta-learning for domain generalization.
- **MetaReg** (Balaji et al., 2018): Metareg: Towards domain generalization using meta-regularization.
- **MASF** (Dou et al., 2019): Domain generalization via model-agnostic learning of semantic features.
- **JiGen** (Carlucci et al., 2019): Domain Generalization by Solving Jigsaw Puzzles.
- **DMG** (Chattopadhyay et al., 2020): Learning to balance specificity and invariance for in and out of domain generalization.
- **L2A-OT** (Zhou et al., 2020a): Learning to Generate Novel Domains for Domain Generalization.
- **RSC** (Huang et al., 2020): Self-challenging improves cross-domain generalization.
- **GroupDRO** (Sagawa et al., 2019): Distributionally robust neural networks for group shifts: On the importance of regularization for worst-case generalization.
- **SFA-A** (Li et al., 2021): A Simple Feature Augmentation for Domain Generalization.
- **MTL** (Blanchard et al., 2021): Domain Generalization by Marginal Transfer Learning.
- **SagNets** (Nam et al., 2021): Reducing Domain Gap by Reducing Style Bias.
- **MatchDG** (Mahajan et al., 2021): Domain generalization using causal matching.

## C.4 IMPLEMENTATION DETAILS

We detail our algorithm's implementation *sub-space Indicator* $\Gamma$: given latent vector $z = g(x) \in \mathbb{R}^{D \times D'}$ (i.e. $z = [z_1, .., z_D]|z_d \in \mathbb{R}^{D'}$), we model $\Gamma(.)$ as a composition of independent indicator on each attribute: $\Gamma(\cdot) = \{\Gamma_d(\cdot)\}_{d=1...D}$ where each $\Gamma_d$ is neural network. In current work, we model $\Gamma_d$ as a Full-Connected (FC) layer $[D' \times 1]$ prior to a sigmoid function $\sigma$ and $\Gamma_d$ takes $z_d$ as input. (Code for experiments is available in Supplementary Material.)

The following are adapted versions on backbones, in which the last FC layer is used as the classification network. The summary of the additional parameter is presented in Table.9.

**AlexNet:** For AlexNet backbone, the extracted feature dimensions are $z \in \mathbb{R}^{4096}$. In current implementation, we partition $z$ into 256 features $z_d \in \mathbb{R}^{16}$ as input for $\Gamma_d$.

**ResNet18 & ResNet50:** For ResNet18 & ResNet50 backbones, instead of partitioning the feature tensor, we utilize direcly feature tensor before avepooling layer. In particular, the extracted feature are $z \in \mathbb{R}^{512 \times 7 \times 7}$ for ResNet18 and $z \in \mathbb{R}^{2048 \times 7 \times 7}$ for ResNet50. We use $z_d \in \mathbb{R}^{7 \times 7 = 49}$ as input for $\Gamma_d$.

**Optimization:** We adopt the SDG optimizer for training with: the model learning-rate ($\text{lr}_{h,g}$) and *sub-space indicator* $\Gamma$ learning-rate ($\text{lr}_\Gamma$) for different bacbones and datasets presented in Table 10, mini-batch size of 64 for 100 epochs. The learning rate ($\text{lr}_{h,g}$ and $\text{lr}_\Gamma$) decayed by 0.1 after every 80% epochs. The values of hyper-parameters $\text{lr}_{h,g}$, $\text{lr}_\Gamma$ and $\tau$ are chosen based on the performance on source validation set. Additionally, in order to the *sub-space indicator* $\Gamma$ quickly adapt to features induced by $h, g$, $\text{lr}_\Gamma$ is set much larger than $\text{lr}_{h,g}$.

Table 9: Additional parameters summary

| Backbones | Partitions | Original | Additional |
|-----------|-----------|----------|------------|
| AlexNet | 256 | 61M | 4352 (0.007%) |
| ResNet18 | 512 | 11M | 25600 (0.230%) |
| ResNet50 | 2048 | 23M | 0.1M (0.440%) |

Table 10: Hyperparameters.

| Dataset | Backbone | $lr_{h,g}$ | $lr_{\Gamma}$ |
|---------|----------|-----------|--------------|
| VLCS | AlexNet | $1.0e^{-3}$ | $1.0e^{-2}$ |
| Office-Home | ResNet18 | $6.5e^{-4}$ | $1.0e^{-0}$ |
|  | ResNet50 | $6.5e^{-4}$ | $1.0e^{-0}$ |
| PACS | AlexNet | $1.0e^{-3}$ | $1.0e^{-2}$ |
|  | ResNet18 | $1.0e^{-2}$ | $1.0e^{-0}$ |
|  | ResNet50 | $1.0e^{-2}$ | $1.0e^{-0}$ |

The number of sampling times in Eq. 8 is also a hyper-parameter in our LASSO. When trying with several possible values of this hyper-parameter, we observed that (i) when increasing the number of sampling times, the performance slightly improves but the training time more significantly increases, and (ii) when setting this hyper-parameter to 1, the performance consistently is satisfactory for all experimental datasets. Therefore, in all experiments, we sample once per mini-batch for computation efficiency. Additionally, at inference time, we conveniently choose $\tau = 0.6$ for ResNet18 & ResNet50 and $\tau = 0.5$ for AlexNet on all datasets. However, from the results in Table 4, it is reasonable to choose $\tau$ based on the performance of the validation data of source domains although sometimes the best performance on validation data does not reflect the best performance on target domain.

**Data augmentation:** We follow the image augmentation protocol introduced in (Gulrajani & Lopez-Paz, 2021) which is increasingly standard in state-of-the-art DG work: crops of random size and aspect ratio, resizing to $224 \times 224$ pixels, random horizontal flips, random color jitter, grayscaling the image with 10% probability, and normalization using the ImageNet channel statistics.

Finally, we use system of GPU NVIDIA Tesla V100 with dual CPUs Intel Xeon E5-2698 v4 to conduct our experiments.

