# OpenReview forum: "LASSO: Latent Sub-spaces Orientation for Domain Generalization"
_ICLR.cc/2022/Conference — ICLR 2022 Submitted_

### Official Review · Reviewer_H9Wv · 2021-10-29

**Correctness:** 2
**Technical Novelty And Significance:** 3
**Empirical Novelty And Significance:** 2
**Recommendation:** 5
**Confidence:** 3

**Main Review:**

Although the paper presents lots of theorems, this reviewer could not find a clear justification for the proposed method.

(1) How the diversity of latent representations could reduce the latent data shift issue:

Theorem 1 and the first paragraph of page 4 suggest that to achieve domain generalization one has to reduce the latent data shift, which is unfortunately not straightforward since no target data is available in the task. The second paragraph argues that learning a single domain-invariant model is not enough to resolve the issue, but such an argument is not theoretically proven. Moreover, in the third paragraph, the authors claim that modeling and utilizing diverse latent representations are the key to the success, but the connection between the diversity of multiple latent representations and the latent data shift issue is totally missing. (Further, considering the technical details in Sec. 3.4, there is no guarantee of the diversity in the proposed method.)

(2) How the sub-space hypothesis could reduce the latent data shift:

In page 5, the authors argue that the compactness of the sub-space can reduce the latent domain shift, which is however not proven theoretically either.

(3) How the mutual information maximization is linked to reducing the latent data shift:

Theorem 3 derives a lower bound of the mutual information between ground-truth label and data projected to the sub-space (i.e., masked feature representation), and accordingly, the masking module is learned to maximize the mutual information in Eq. (6). However, there is no justification for this approach nor the connection between the mutual information and the latent data shift (or domain generalization in general).

(4) How the independence of the masking weights leads to better generalization:

The argument in page 6 about the advantages of such independence is not rigorous; the reason following the argument seems indecipherable.

Due to these, this reviewer feels the theorems derived in this paper are not very useful to justify the proposed method, and cannot find clear reasons for its advantages.


[Other comments]
- Do not understand why z is a 2D tensor and what each dimension stands for.
- Would recommend drawing a figure to better illustrate the overall pipeline.
- The first theorem resembles the well known theorem in literature of domain adaptation. It would be useful if their differences are well explained in the paper.


**Summary Of The Paper:**

This paper introduces a new method for domain generalizable classification. Given a feature representation of input, the proposed method first samples binary masks from a Bernoulli distribution whose parameters are determined by the feature, projects the input feature to multiple sub-spaces by using the masks, and applying classifiers associated with each of the sub-spaces. The parameters of the Bernoulli distribution are predicted by a module that is learned together with the feature representation and classifiers.

Although the paper tries to provide mathematical backgrounds and underlying theories of the proposed method, unfortunately, this reviewer finds no clear justification for learning and using the sub-spaces. Also, the improvement by the method seems marginal (or it even underperforms sometimes) when coupled with small backbone networks.


**Summary Of The Review:**

This reviewer feels the theorems derived in this paper are not very useful to justify the proposed method, and cannot find clear reasons for its advantages. Also, the practical value of the method seems limited as its performance is not impressive or even inferior to previous work when it is incorporated with small networks.

-- post-rebuttal --
I greatly appreciate the kind responses and the update of the manuscript and would like to upgrade my rating accordingly, but since my major concern has not been resolved yet, I am still leaning towards rejection.

---

> ### Author Response · Authors · 2021-11-19
> **Response to Reviewer H9Wv (1/2)**
>
> We thank the reviewers for your constructive comments. We hope that we can address some of your points as presented below.
>
> &nbsp;
>
> ### **Q1: How the sub-space hypothesis could reduce the latent data shift**
>
> Given a latent representation $z$ with a label $y$ in the high-dimensional latent space, only a small portion of its features known as *label-informative features* is highly relevant to the label $y$, while the remaining ones are redundant. This  observation is supported by our ablation study about the effect of $\tau$ which is presented in Table 4 (main paper).
>
> By eliminating irrelevant features and grouping the latent representations of data examples across multiple domains with the same set of label-informative features, we can form latent label-informative sub-spaces to reduce the latent data shift, whereas preserving *sufficient label-information* for training *good hypotheses* on those *latent sub-spaces*.  This can be explained from the fact that unseen target and source examples with the same label-informative features are projected onto the same latent sub-space on which the latent data shift between mixture of source domains and target domain becomes smaller due to the compactness of this sub-space compared with the full high-dimensional latent space.
>
> From our ablation study about sub-space representation, is can be seen that the latent is more compact and easy to classify on sub-space. The numerical results also demonstrate that the latent data shift can be reduced on sup-space.
>
> &nbsp;
>
> ### **Q2: How the diversity of latent representations and appropriate hypotheses improve generalization**
>
> By learning multiple hypotheses, each of which corresponds to a sub-space, LASSO allows sub-spaces to instance-wisely explore different groups of label-informative features, hence encouraging the diversity of latent representations for achieving better generalization ability, which concurs with the principle in (Huang et al., 2020; Chattopadhyay et al., 2020; Blanchard et al., 2021).
>
> The reason is that given source examples with their labels, LASSO aims to explore possible compact groups of label-informative features which can predict accurately the labels. In the inference time, given a target example, if the feature extractor can successfully activate a group of label-informative features, this target example is projected and matched with corresponding source examples on a sub-space in which a good hypothesis is used to predict a label for the target example.
>
> **Update:** Following the suggestion of the reviewer, we have revised discussion after Theorem 1 to further clarify motivation and added Figure 1 with brief description to better illustrate the overall idea.
>
> &nbsp;
>
> ### **Q3: How the mutual information maximization is linked to reducing the latent data shift**
> It is evident that if the sub-space indicator $\Gamma$ can select and retain qualified label-informative features, source and target examples are projected onto compact sub-spaces, while preserving sufficient label information to train qualified sub-space hypotheses. Due to the compactness of the sub-spaces, the latent data shifts in sub-spaces are smaller that that of the full high-dimensional space. Additionally, our ablation study in Section 4.2 validates this observation.
>
> Theorem 3 states that by solving the optimization problem of LASSO, we implicitly maximize the mutual information of selected features by $\Gamma$ and labels. By this way, the sub-space indicator $\Gamma$ and feature extractor $g$ in LASSO are in cooperation to retain the features that contain most information of the labels.

---

> > ### Author Response · Authors · 2021-11-19
> > **Response to Reviewer H9Wv (2/2)**
> >
> >
> > &nbsp;
> >
> > ### **Q4: How the independence of the masking weights leads to better generalization**
> >
> > **2D tensor**: We decomposed feature vector $z$ into many sub-groups i.e., $z=\left [z_{1},...,z_{D}\right ]\in\mathbb{R}^{D'\times D}$ whereas each $z_{d}\in\mathbb{R}^{D'}$ known as **an attribute** consists of $D'$ features.
> > For example:
> > - AlexNet:  the extracted feature dimensions are $z\in \mathbb{R}^{4096}$. In current implementation, we partition $z$ into 256 attributes $z_d \in \mathbb{R}^{16}$ as input for $\Gamma_d$.
> > - ResNet18 \& ResNet50: instead of partitioning the feature tensor, we utilize direcly feature tensor before avepooling layer. In particular, the extracted feature are $z\in \mathbb{R}^{512\times 7\times 7}$ for ResNet18 and $z\in \mathbb{R}^{2048\times 7\times 7}$ for ResNet50. We use $z_d \in \mathbb{R}^{7\times 7=49}$ as input for $\Gamma_d$.
> >
> >
> >
> > **Attribute-based $\Gamma$**: we model sub-space indicator as $\Gamma\left(z\right)=\left[\Gamma_{1}\left(z_{1}\right),...,\Gamma_{D}\left(z_{D}\right)\right]\in\left[0,1\right]^{D}$. Consequently, $\Gamma_{d}\left(z\right)=\Gamma_{d}\left(z_{d}\right)$ is computed based solely on the group of $D'$ features in the attribute $z_{d}$ itself rather than full latent $z$. That means attribute-based $\Gamma_d$ depends on attributes which are shared across domains instead of domains, hence, becomes more independent from domain information. Moreover, by Theorem 3, model can still learn meaningful attributes as long as the performance on source domains is guaranteed. In unseen domains, the target of sub-space indicator $\Gamma$ is to detect label-informative attributes *which are learned in source domains* instead of identifying all label-informative attributes.
> >
> > More specifically, looking at each $\Gamma_{d}\left(z_{d}\right)$ in detail, since $z_d$ come from different source domains, the corresponding $\Gamma_d$ plays as a domain invariant hypothesis which output {0, 1} identifying label-informative attributes across source domains. Therefore, in target domain, $\Gamma_{d}\left(z_{d}\right)$ only predict $z_d$ is label-informative attribute *which is learned in source domains* or not.
> >
> > Our intuition is supported by ablation study about the effect of the attribute-based indicator $\Gamma_d$ which is presented in Table 5, Appendix B.1.
> >
> > &nbsp;
> >
> >
> > **Remark:** We would like to highlight that our Theorem 1 despite developing in a general context of DG is also novel due to two following reasons: (i) it proposes the target loss upper-bound in a general setting of  multi-class classification with a sufficiently general loss, which can be viewed as a non-trivial generalization of existing works in DA and DG [1,2,3] and (ii) our  bound is interweaving both input and latent spaces, which is novel and appropriate for theoretical analyses in deep learning,  while the bounds in previous works only involve input space. We have clarified the novelty in the Appendix A.
> >
> > &nbsp;
> >
> >
> > [1] Mansour, Yishay ; Mohri, Mehryar ; Rostamizadeh, Afshin: Domain Adaptation: Learning Bounds and Algorithms.. In: COLT, 2009
> >
> > [2] Ben-David, Shai, et al. "A theory of learning from different domains." Machine learning 79.1 (2010): 151-175.
> >
> > [3] Zhao, Han, et al. "On learning invariant representations for domain adaptation." International Conference on Machine Learning. PMLR, 2019.

---

> > > ### Author Response · Authors · 2021-11-29
> > > **Looking forward to Reviewer's feedback**
> > >
> > > Dear Reviewer,
> > >
> > > Thanks again for your valuable comments and reviews. The discussion period is coming to a close, and we will not be able to directly engage in further discussions after that point. We would really appreciate a reply as to whether our response and clarifications have addressed the issues raised in your review or anything else we can address.
> > >
> > > Authors

---

> > > > ### Comment · Reviewer_H9Wv · 2021-11-30
> > > > **Many thanks for the kind responses - Post rebuttal review**
> > > >
> > > > Thank the authors greatly for their kind responses. Clarifications in the rebuttal and Figure 1 of the revision are useful. Also, I read the paper again and noticed that I misunderstood the independence of the probabilistic sub-space indicator in Section 3.4.1, thus the 4th concern of mine in the review is resolved.
> > > >
> > > > However, I still have a concern over the key assumption that a lower dimensional (i.e., compact) and label-informative subspace guarantees reduced domain shift. I feel the main idea of this paper relies heavily on this assumption, and if the assumption is not true, the theorems in the paper seems not very useful to support the proposed method (I agree with the authors that the theorems could be valuable as-is for domain adaptation and generalization though). Although the assumption is validated empirically and Figure 1 is somehow useful to understand its underlying motivation, its theoretical background is provided neither in the paper nor in the rebuttal. Also, I wonder how the probabilistic subspace indicator could generalize well to examples of unseen domains. In the idea case, the indicator may learn to reveal domain-invariant and discriminative features in an instance-wise manner so that the compact subspace determined by the indicator could reduce the domain shift while retaining high classification accuracy, but I have trouble understanding how the loss in Eq. (7) enables this.
> > > >
> > > > In summary, I greatly appreciate the kind responses and the update of the manuscript and would like to upgrade my rating accordingly, but since my major concern has not been resolved yet, I am still leaning towards rejection.

---

> > > > > ### Author Response · Authors · 2021-12-03
> > > > > **Many thanks for your response and score update**
> > > > >
> > > > > Thanks for your response and score update. We really appreciate this.
> > > > >
> > > > > Regarding the theoretical part, to the best of our knowledge, different from the domain adaptation, there is currently no rigorous theory for domain generalization to tightly bound the loss on unseen target domains due to the great variance of unseen target domains. The general assumption of unseen target domains generated from the same random process (i.e., a distribution over distribution) seems to be overly general and insufficient to motivate more rigorous theory. Most of work in domain generalization needs to make some assumption to go further.
> > > > >
> > > > > Moreover, many works aim to learn domain-invariant features on source domains. From conceptual viewpoint, these works simply borrow the principle of learning domain-invariant features from domain adaptation, but did not mention to how the domain-invariant features generalize to unseen target domains or how to quantify the target loss over unseen target domains caused by the domain-invariant features on source domains.
> > > > >
> > > > > Comparing to these works, with the assistance of Theorem 1 in our work, we can realize the factors influencing the loss on unseen target domains. We regret that to motivate from *single-space & single hypothesis* to *multiple-spaces & multiple-hypotheses*, we need to rely on the intuition and motivation that when grouping data examples by label-informative features and projecting them to these sub-spaces, we can reduce the domain shift, while preserving sufficient label information to learn good sub-space hypotheses.
> > > > >
> > > > >
> > > > > Actually, this can be partly confirmed from the monotonicity of a general $f-divergence$ for which we have: $D_f(P_{X,Y} || Q_{X,Y}) \geq D_f(P_{X} || Q_{X})$ which implies that projecting to a sub-space helps to reduce $f-divergence$.
> > > > >
> > > > > However, if we cannot choose a good set of features $X$, we cannot train good hypothesis on this sub-space. Our Theorem 3 links to information theory to say that by minimizing the losses of hypotheses on sub-spaces, we can maximize the mutual information $I(\Gamma(g(X)) \odot X, g(X), Y)$ to enable the sub-space indicator $\Gamma$ to learn good and label-informative features.
> > > > >
> > > > > Regarding your concern about the loss in Eq. (7), as shown in Eq.(6), this loss is indeed the total loss of sub-space hypotheses. By Theorem 3, we prove that by minimizing this loss, we maximize the mutual information $I(\Gamma(g(X)) \odot X, g(X), Y)$ to enable learning good and label-informative features.
> > > > > .

---

### Official Review · Reviewer_gyy1 · 2021-11-01

**Correctness:** 3
**Technical Novelty And Significance:** 2
**Empirical Novelty And Significance:** 2
**Recommendation:** 6
**Confidence:** 5

**Details Of Ethics Concerns:**

See the above Main Review Section.

**Main Review:**

Instead of using a single consistent latent space, the authors propose to employ multiple latent spaces by capturing the label-informative features from the available source domains. Some theoretical analyses are given to support their learning scheme, and a very simple latent subspace orientation algorithm is proposed to tackle the domain generalization challenges. Experiments validate the usefulness of the proposed method on different datasets.

Some concerns should be considered for rebuttal.
1. Generally, the proposed method alternatively optimizes an indicator function and subspace learning. Although there are three theorems to support your claim, the reviewer still has concerns about the standpoint and update of the subspace indicator. Apart from these claimed theoretic analyses on the subspace indicator \Delta, the authors should give a more intuitive explanation of why it is useful, which can help the readers understand your work, rather than lots of equations. Moreover, the reviewer is wondering if we fix these indicator updates after some iterations, what will happen for the performance because the review doubt such an update is meaningless for this task in practical systems.
2. Why is maximizing the mutual information useful in your framework? From the reviewer's view, using other adversarial learning or metric learning also can achieve your objective.
3. The presented performance is unsatisfactory when compared to other works published in 2021, although this paper is very simple and easy to implement.
4. The authors should clarify why the performance of using Threshold outperforms those of using ensemble. Intuitively, using an ensembling strategy has a tendency to have better performance. Moreover, how could you set the value of \tau? Is it a hyper-parameter? Is there any adaptive learning strategy to determine it or empirically setting?
5. Many grammar mistakes are shown in the work, and the authors should pay attention to your presentation.


**Summary Of The Paper:**

This paper proposes a latent subspace orientation algorithm for domain generalization, which is built on diverse latent subspaces and individual hypotheses with label-informative features. Extensive theoretic and experiments show the effectiveness of the proposed method with two variants.

**Summary Of The Review:**

This paper proposes a simple latent subspace orientation algorithm for domain generalization based on some theoretical and experimental observations. The technical parts with some complex theoretical analyses are satisfied, yet the model construction and experimental parts could be improved.

---

> ### Author Response · Authors · 2021-11-19
> **Response to Reviewer gyy1 (1/2)**
>
> Thank you very much for your constructive comments. We hope that we can address some of your points as presented below.
>
> &nbsp;
>
> ### **Motivation and Intuition of Sup-space**
>
> Given a latent representation $z$ with a label $y$ in the high-dimensional latent space, only a small portion of its features known as *label-informative features* is highly relevant to the label $y$, while the remaining ones are redundant. This observation is supported by our ablation study about the effect of $\tau$ which is presented in Table 4 (main paper).
>
> By eliminating irrelevant features and grouping the latent representations of data examples across multiple domains with the same set of label-informative features, we can form latent label-informative sub-spaces to reduce the latent data shift, whereas preserving *sufficient label-information* for training *good hypotheses* on those *latent sub-spaces*.  This can be explained from the fact that unseen target and source examples with the same label-informative features are projected onto the same latent sub-space on which the latent data shift between mixture of source domains and target domain becomes smaller due to the compactness of this sub-space compared with the full high-dimensional latent space.
> From our ablation study about sub-space representation, is can be seen that the latent is more compact and easy to classify on sub-space. The numerical results also demonstrate that the latent data shift can be reduced on sup-space.
>
> In addition, by learning multiple hypotheses, each of which corresponds to a sub-space, LASSO allows sub-spaces to instance-wisely explore different groups of label-informative features, hence encouraging the diversity of latent representations for achieving better generalization ability, which concurs with the principle in (Huang et al., 2020; Chattopadhyay et al., 2020; Blanchard et al., 2021).
>
> The reason is that given source examples with their labels, LASSO aims to explore possible compact groups of label-informative features which can predict accurately the labels. In the inference time, given a target example, if the feature extractor can successfully activate a group of label-informative features, this target example is projected and matched with corresponding source examples on a sub-space in which a good hypothesis is used to predict a label for the target example.
>
> **Update:** Following the suggestion of the reviewer, we have revised discussion after Theorem 1 to further clarify motivation and added Figure 1 with brief description to better illustrate the overall idea.
>
> &nbsp;
>
> ### **Mutual information**
>
> In our work, we do not explicitly maximize the mutual information. Through Theorem 3, we theoretically demonstrate that by learning a sub-space indicator $\Gamma$ supporting the sub-space hypotheses to predict accurately data examples in the sub-spaces, we can implicitly maximize the relevant mutual information.
>
> Theorem 3 also indicates that the optimization problem of LASSO enables learning the sub-space indicator $\Gamma$ and feature extractor $g$ to maximize the mutual information $\mathbb{I}\left(\Gamma\left(g(X)\right)\odot g\left(X\right),Y\right)$,
> which enables the *sub-space indicator $\Gamma$* to preserve the label-informative features of $z=g\left(x\right)$.
>
> Therefore, in Theorem 3, we theoretically connect our theory with mutual information, which guarantees the rationality of the sub-space indicator $\Gamma$ in choosing label informative and relevant features.
>
> Moreover, from the results in Table 4 (main paper),
> it can be seen that when we increase the confidence of *sub-indicator $\Gamma$* (increase value of threshold $\tau$), the model utilizes only a smaller number of features  but nonetheless still yields good results. This indicates that *sub-indicator $\Gamma$* is able to identify essential label-informative features.

---

> > ### Author Response · Authors · 2021-11-19
> > **Response to Reviewer gyy1 (2/2)**
> >
> > &nbsp;
> >
> > ### **Inference with "threshold" and "ensemble", and How to Set $\tau$**
> > Intuitively, the threshold approach with a threshold $\tau$ selects and retains the top most relevant and label-informative features, while the ensemble approach aggregates the predictions of possible sub-space hypotheses. We conjecture that the slight superior of the threshold approach to the ensemble one is possibly due to the fact that the former with an appropriate $\tau$ can select and retain more compact and label-informative sets of features.  Moreover, the threshold $\tau$ can be set conveniently, e.g., we use $\tau= 0.6$ for ResNet18 & ResNet50 and $\tau= 0.5$ for AlexNet on all datasets.
> >
> > Furthermore, the fact that the threshold and ensemble approaches have obtained the good and close performances indicates the sub-space indicator $\Gamma$ can really select and retain qualified label-informative features so that regardless we use a threshold to select label-informative features or sample them, we obtain quite similar performances.
> > From the results in Table 4 (in main paper), it is reasonable to choose $\tau$ based on the performance of the validation data of source domains although sometimes the best performance on validation data does not reflect the best performance on target domain.
> >
> > &nbsp;
> >
> > ### **Comparison to baselines in 2021**
> > We have added some works published in 2021 to the benchmarks including [1] (CVPR 2021), [2] (NeurIPS 2021), and [3] (ICML 2021). According to Tables 1 and 2, our LASSO is the best in both Table 1 for OfficeHome and Table 2 for PACS.
> >
> > In addition, from the methodology perspective, compared to the above baselines, as you commented, our LASSO is much simpler and easier to implement with only one easy to set parameter. We believe that in practice, simpler methods seem to be favored due to their convenient and comfortable deployment.
> >
> > ***Table 1: Classification Accuracy on Office-Home.***
> >
> > | Method | Backbone | Art | Clipart | Product | RealWorld | Average |
> > | :---- | :----: | :---- | :---- | :---- | :---- |----: |
> > SagNets [1] | ResNet50 | 63.40 +/- 0.20  | 54.80 +/- 0.40  | 75.80 +/- 0.40  | 78.30 +/- 0.30 | 68.10 |
> > mDSDI [2] | ResNet50 | 68.10 +/- 0.30  | 52.10 +/- 0.40 | 76.00 +/- 0.20 |  80.40 +/- 0.20 |  69.20 |
> > LASSO-$\tau$ | ResNet50 | 66.56 +/- 0.04 | 58.21 +/- 0.19 | 78.69 +/- 0.01 | 79.27 +/- 0.42 | **70.70** |
> > LASSO-En | ResNet50 | 67.07 +/- 0.38 | 58.13 +/- 0.31 | 78.27 +/- 0.02 | 79.19 +/- 0.41 | 70.66 |
> >
> > &nbsp;
> >
> > ***Table 2: Classification Accuracy on PACS.***
> >
> > | Method | Backbone | Photo | Art-painting | Cartoon | Sketch | Average |
> > | :---- | :----: | :---- | :---- | :---- | :---- |----: |
> > SagNets [1] | ResNet50 | 97.10 +/- 0.10 | 87.40 +/- 1.00 | 80.70 +/- 0.60 | 80.00 +/- 0.40 | 86.30 |
> > mDSDI [2] | ResNet50 | 98.10 +/- 0.30 | 87.70 +/- 0.40 | 80.40 +/- 0.70 |  78.40 +/- 1.20 | 86.20 |
> > MatchDG [3] | ResNet50 |  97.94 +/- 0.00 | 85.61 +/- 0.0 | 82.12 +/- 0.00 | 78.76 +/- 0.00 | 86.11 |
> > LASSO-$\tau$ | ResNet50 | 96.62 +/- 0.56 | 87.81 +/- 0.69 | 82.99 +/- 0.71 | 82.33 +/- 0.20 | **87.43** |
> > LASSO-En | ResNet50 | 96.93 +/- 0.65 | 87.23 +/- 0.57 | 82.55 +/- 1.09 | 81.95 +/- 0.37 | 87.15 |
> >
> > &nbsp;
> >
> > **Grammar mistakes:** Thanks for this comment. We have done proof-reading the paper to fix grammar mistakes.
> >
> > &nbsp;
> >
> > [1] Nam, Hyeonseob, et al. "Reducing Domain Gap by Reducing Style Bias." Proceedings of the IEEE/CVF Conference on Computer Vision and Pattern Recognition. 2021.
> >
> > [2] Bui, Manh-Ha, et al. "Exploiting Domain-Specific Features to Enhance Domain Generalization." Advances in Neural Information Processing Systems 34 (2021).
> >
> > [3] Mahajan, D., Tople, S., & Sharma, A. (2021, July). Domain generalization using causal matching. In International Conference on Machine Learning (pp. 7313-7324). PMLR.

---

> > > ### Author Response · Authors · 2021-11-29
> > > **Looking forward to Reviewer's feedback**
> > >
> > > Dear Reviewer,
> > >
> > > Thanks again for your valuable comments and reviews. The discussion period is coming to a close, and we will not be able to directly engage in further discussions after that point. We would really appreciate a reply as to whether our response and clarifications have addressed the issues raised in your review or anything else we can address.
> > >
> > > Authors

---

> > > > ### Comment · Reviewer_gyy1 · 2021-12-03
> > > > **Thanks for post-rebuttal**
> > > >
> > > > The authors almost clarified their claims and my concerns. After reading the discussion and rebuttal from other reviewers, I would like to raise my score to borderline accept.

---

> > > > > ### Author Response · Authors · 2021-12-05
> > > > > **Thanks for your score update**
> > > > >
> > > > > Many thanks for your score update. We really appreciate it.

---

### Official Review · Reviewer_aksw · 2021-11-01

**Correctness:** 3
**Technical Novelty And Significance:** 3
**Empirical Novelty And Significance:** Not applicable
**Recommendation:** 5
**Confidence:** 4

**Main Review:**

In the abstract, the author claimed that exploring diverse latent sub-spaces for individual hypotheses learning is superior compared with learning single hypothesis shared among domains, however, it is not clear to me what is the significance of doing so.  The authors try to give an explanation by delivering an upper bound target general loss, which to me is quite questionable since the target domain is not available during the training phase. Simply using the mixture sources domain may not reflect the true distribution of the target one.
Besides there are some other weaknesses as follows:
1. Is Theorem 2 strictly lower than Theorem 1? I can only find the author’s explanation rather than strict prove. This part is essential for this paper, besides, this is closely related to my given score.
2. In Section3.4.2, the author proposes to use sampling for model training. I wonder about the performance of doing this in a deterministic manner, just like using threshold at the testing phase.
3. Experiments are somewhat lacking, 1) challenging datasets (e.g., medical imaging for CV, reinforcement learning in MLDG) should be considered since the proposed method is more related with general machine learning,  2) only RSC is considered as the latest stoa baseline, which is not desired.

Minor:
4. For the experimental part, why not keep the same manner to select the validation set? It would be better to keep this unchanged.


**Summary Of The Paper:**

This paper critically re-examines the rationality of domain-invariant based method for DG (generalize to unseen domains without re-training). It highlights that, tackling all source domains equally without taking the underlying relationship between them to learn domain-invariant features can lead to a sub-optimal solution. To deal with this, it introduces a latent space decoupling-based method called Latent Sub-Space Orientation (LASSO) to eliminate some irrelative representations for prediction. By this, the relationship between latent space and label space can be strengthened, thus the model can generalize better. The results show that LASSO (especially with threshold) can commonly lead to a marginal performance boost for popular DG benchmarks.

**Summary Of The Review:**

Though I am not an expert in this area, I have discussed my comments with my colleagues with rich experience in this area. Thus, I am still confident in my comments.

---

> ### Author Response · Authors · 2021-11-19
> **Response to Reviewer aksw (1/2)**
>
> Thank you very much for your constructive comments. We hope that we can address some of your points as presented below.
>
> ### **Theorem 1 and Theorem 2**
> Theorem 2 is an extension of Theorem 1 in the context of multiple sub-spaces and multiple hypotheses. Theorem 1 is developed for the case of single space and single hypothesis, while Theorem 2 is developed for the case multiple sub-spaces and multiple hypotheses. We do not claim that the bound in Theorem 2 is strictly lower than that in Theorem 1 because it is hard to verify rigorously from the theoretical perspective.
>
> Indeed, our pathway is as follows. We establish Theorem 1 to see the factors really influencing the performance on unseen target domains. From that we motivate our multiple sub-spaces and hypotheses approach, and establish Theorem 2 for our approach to see the factors really influencing the performance on unseen target domains. Finally, we develop Theorem 3 to justify the rationale of the sub-space indicator $\Gamma$ in selecting and retaining most label-informative features. In what follows, we would like to detail our pathway and motivation.
>
> From Theorem 1, it can be seen that the latent data shift between the mixture of source domains and unseen target domains is one of key factors to reduce the loss on unseen target domain. However, in the context of domain generalization, this task is very challenging due to unseen target domains.
> A large number of works propose learning domain-invariant features on a full high-dimensional latent space together with a single hypothesis on top of these domain-invariant features. Nonetheless, due to the great variance of unseen target distributions on the full high-dimensional latent space, the latent data shift is possibly high in many cases, which hurts the generalization ability of the single hypothesis on unseen target domains.
>
> Evidently, given a latent representation $z$ with a label $y$ in the high-dimensional latent space, only a small portion of its features known as *label-informative features* is highly relevant to the label $y$, while the remaining ones are redundant. This  observation is supported by our ablation study about the effect of $\tau$ which is presented in Table 4 (main paper).
>
> By eliminating irrelevant features and grouping the latent representations of data examples across multiple domains with the same set of label-informative features, we can form latent label-informative sub-spaces to reduce the latent data shift, whereas preserving *sufficient label-information* for training *good hypotheses* on those *latent sub-spaces*.  This can be explained from the fact that unseen target and source examples with the same label-informative features are projected onto the same latent sub-space on which the latent data shift between mixture of source domains and target domain becomes smaller due to the compactness of this sub-space compared with the full high-dimensional latent space.
> From our ablation study about sub-space representation, is can be seen that the latent is more compact and easy to classify on sub-space. The numerical results also demonstrate that the latent data shift can be reduced on sup-space.
>
> In addition, by learning multiple hypotheses, each of which corresponds to a sub-space, LASSO allows sub-spaces to instance-wisely explore different groups of label-informative features, hence encouraging the diversity of latent representations for achieving better generalization ability, which concurs with the principle in (Huang et al., 2020; Chattopadhyay et al., 2020; Blanchard et al., 2021).
>
> The reason is that given source examples with their labels, LASSO aims to explore possible compact groups of label-informative features which can predict accurately the labels. In the inference time, given a target example, if the feature extractor can successfully activate a group of label-informative features, this target example is projected and matched with corresponding source examples on a sub-space in which a good hypothesis is used to predict a label for the target example.

---

> > ### Author Response · Authors · 2021-11-19
> > **Response to Reviewer aksw (2/2)**
> >
> > &nbsp;
> >
> > The above discussions motivate our LASSO with multiple sub-spaces and multiple hypotheses. We then develop Theorem 2 to inspect the factors which influence performances on unseen target domains. It can be seen that the key factor of our approach is that the sub-space indicator $\Gamma$ needs to be able to select and retain qualified label-informative features. We then develop Theorem 3 to theoretically confirm that our LASSO with its optimization problem can maximize the mutual information of selected features and labels, hence can select and retain qualified label-informative features.
> >
> > If we can learn a qualified *sub-space indicator $\Gamma$*, we can project corresponding latent representations onto low-dimensional sub-spaces conducted from label-informative features that reduces the latent data shift on the sub-spaces due to the compression effect when projecting to the sub-spaces. Moreover, if we can appropriately choose the good sets of label-informative features for the sub-spaces, we can preserve sufficient label information on the sub-spaces for training good hypotheses $f_m$ or $h_m$ with a low  source sub-space hypothesis loss $\mathcal{L}\left(f_{m},\mathbb{D}_{m}^{S}\right)$ with the aim to lower the upper-bound.
> >
> > **Update:** We have revised discussion after Theorem 1 to clear our motivation and added Figure 1 with brief description to better illustrate the overall idea.
> >
> > &nbsp;
> >
> > ### **Sampling for model training**
> > In training phase, sampling $m\sim Ber\left(\Gamma\left(z\right)\right)$ would help hypotheses explore more useful sub-spaces. The motivation is that target domains might have or share a small sub-set of learned features in source domains, e.g., in PACS dataset, many label-informative features can be learned in rich information domain such as "Photo" domain while "Sketch" domain only has edge/shape information for classification.
> > Therefore, in training phase,
> > we would like to train hypotheses on different sub-set of learned features (i.e. sub-spaces of sub-space) which would strengthen sub-space hypotheses on arbitrary target domains..
> >
> > ***Table 1: Classification Accuracy on PACS.***
> >
> > | Method | Backbone | Photo | Art-painting | Cartoon | Sketch | Average |
> > | :---- | :----: | :---- | :---- | :---- | :---- |----: |
> > Deterministic-$\tau$ | ResNet50 | 95.99 +/- 0.42 | 86.28 +/- 1.70 | 80.09 +/- 1.54 | 79.01 +/- 4.44 | 85.34 |
> > Deterministic-En | ResNet50 | 93.05 +/- 0.03 | 86.18 +/- 1.08 | 80.08 +/- 1.62 | 77.73 +/- 2.65 | 84.26 |
> > LASSO-$\tau$ | ResNet50 | 96.62 +/- 0.56 | 87.81 +/- 0.69 | 82.99 +/- 0.71 | 82.33 +/- 0.20 | **87.43** |
> > LASSO-En | ResNet50 | 96.93 +/- 0.65 | 87.23 +/- 0.57 | 82.55 +/- 1.09 | 81.95 +/- 0.37 | *87.15* |
> >
> > To demonstrate the benefit of sampling, we conduct additional experiments using deterministic manner which uses threshold $\tau$ to determine mask $m$ in model training. The results in Table 1 indicate sampling significantly enhance performance.
> >
> >
> >
> > We have added additional experiments about sampling for model training to the Appendix B.2.

---

> > > ### Author Response · Authors · 2021-11-29
> > > **Looking forward to Reviewer's feedback**
> > >
> > > Dear Reviewer,
> > >
> > > Thanks again for your valuable comments and reviews. The discussion period is coming to a close, and we will not be able to directly engage in further discussions after that point. We would really appreciate a reply as to whether our response and clarifications have addressed the issues raised in your review or anything else we can address.
> > >
> > > Authors

---

> > > > ### Comment · Reviewer_aksw · 2021-12-03
> > > > **My feedback for the authors' response**
> > > >
> > > > I feel sorry for my late reply since I am busy with some personal things recently. Thanks for the response. I am still not convinced by the theorem part. The authors mentioned that Th 1 is to see the factors really influencing the performance on the unseen target domain, which to me is not feasible. Do you have any assumptions about the target domain? If not, IMHO, it is impossible to evaluate/obtain a risk bound on the unseen target domain. Thanks for the response for the sampling part, my concern about this has been addressed. For the experiment, I didn’t find evaluation on challenging DG tasks, which makes the significance of the proposed method somewhat lacking. But I think it is acceptable for fundamental machine learning research.
> > > > To sum up, I will keep my score unchanged at this stage, mainly due to the theorem part.

---

> > > > > ### Author Response · Authors · 2021-12-03
> > > > > **Thanks for your feedback**
> > > > >
> > > > > Thanks for your feedback. We really appreciate this.
> > > > >
> > > > > Regarding the theoretical part, to the best of our knowledge, different from the domain adaptation, there is currently no rigorous theory for domain generalization to tightly bound the loss on unseen target domains due to the great variance of unseen target domains. The general assumption of unseen target domains generated from the same random process (i.e., a distribution over distribution) seems to be overly general and insufficient to motivate more rigorous theory. Most of work in domain generalization needs to make some assumption to go further.
> > > > >
> > > > > Furthermore, many works aim to learn domain-invariant features on source domains. From conceptual viewpoint, these works simply borrow the principle of learning domain-invariant features from domain adaptation, but did not mention to how the domain-invariant features generalize to unseen target domains or how to quantify the target loss over unseen target domains caused by the domain-invariant features on source domains.
> > > > >
> > > > > Comparing to these works, with the assistance of Theorem 1 in our work, we can realize the factors influencing the loss on unseen target domains. We regret that to motivate from *single-space & single hypothesis* to *multiple-spaces & multiple-hypotheses*, we need to rely on the intuition and motivation that when grouping data examples by label-informative features and projecting them to these sub-spaces, we can reduce the domain shift, while preserving sufficient label information to learn good sub-space hypotheses.
> > > > >
> > > > > Actually, this can be partly confirmed from the monotonicity of a general $f-divergence$ for which we have: $D_f(P_{X,Y} || Q_{X,Y}) \geq D_f(P_{X} || Q_{X})$ which implies that projecting to a sub-space helps to reduce $f-divergence$.
> > > > >
> > > > > However, if we cannot choose a good set of features $X$, we cannot train good hypothesis on this sub-space. Our Theorem 3 links to information theory stating by minimizing the losses of hypotheses on sub-spaces, we can maximize the mutual information $I(\Gamma(g(X)) \odot X, g(X), Y)$ to enable the sub-space indicator $\Gamma$ to learn good and label-informative features.

---

> > > > > > ### Comment · Reviewer_aksw · 2021-12-05
> > > > > > **Thanks**
> > > > > >
> > > > > > Thanks for the clarification. It seems that I have some misunderstandings from the previous discussion. After reading this response, I turn to accept this paper for this work. But, I think the authors should better discusses those in the final version of this work.

---

> > > > > > > ### Author Response · Authors · 2021-12-06
> > > > > > > **Looking forward to your score update!**
> > > > > > >
> > > > > > > Many thanks for your reconsideration. We really appreciate it and will definitely update those discussions in the final version of this work.
> > > > > > >
> > > > > > > If it is not too much trouble to ask, could you please also update the score in the original review to reflect your current positive decision?
> > > > > > >
> > > > > > > Best regards,
> > > > > > >
> > > > > > > Authors

---

### Official Review · Reviewer_crRo · 2021-11-03

**Correctness:** 4
**Technical Novelty And Significance:** 3
**Empirical Novelty And Significance:** 3
**Recommendation:** 6
**Confidence:** 4

**Main Review:**

1) Authors correctly point out overly strict and sub-optimal assumption that is popular in the literature currently.

2) Authors’ claims are backed both by theory and experimental evidence.

3) Some benchmark papers in the literature (which don’t have an assumption of domain-invariant features) are not considered [1-2]. Both these papers don’t have an assumption of domain-invariant features and are similar in spirit as the current paper.

4) Theoretical analysis in the given paper helps to come up with a loss but not actually analyze the method. Like in [1-2], can authors comment on the learning theoretic study? How does risk bound change with the number of available domains and number of examples in the training?

5) As we have a large number of domains and a large number of training examples then one should be able to have optimal error or loss.  I could not draw this conclusion from the theory that the authors provided. Can authors comment more on this?

6) How were hyperparameters tuned for the baseline method?

[1] Blanchard, Gilles, Gyemin Lee, and Clayton Scott. "Generalizing from several related classification tasks to a new unlabeled sample." Advances in neural information processing systems 24 (2011): 2178-2186.

[2] Blanchard, Gilles, Aniket Anand Deshmukh, Ürün Dogan, Gyemin Lee, and Clayton Scott. "Domain Generalization by Marginal Transfer Learning." arXiv preprint arXiv:1711.07910 (2017) J. Mach. Learn. Res. (JMLR) 22 (2021): 2-1.


**Summary Of The Paper:**


Authors point out that the assumption of existence of fixed domain-invariant features and common hypotheses learned from a set of training domains could be overly strict and sub-optimal. Authors propose a new method which doesn’t have a single hypothesis shared among domains and give theoretical analysis of the proposed method. Authors also give results on benchmark datasets.


**Summary Of The Review:**

Paper has rigorous set of experiments and is backed by clearly explained motivation. Addition of suggested literature could further improve the paper.

---

> ### Author Response · Authors · 2021-11-20
> **Response to Reviewer crRo**
>
>
> Thank you very much for your possitive comments. We hope that we can address some of your points as presented below.
>
> &nbsp;
>
> **Q1**: Some benchmark papers in the literature  (which don’t have an assumption of domain-invariant features) are not considered [1-2].  Both these papers don’t have an assumption of domain-invariant features and are similar in spirit to the current paper.
>
> Thank you for the suggestion, and we will update our paper to include these papers. From our point of view, the work of Blanchard et al. considers the most general DG setting, in which a meta-distribution generates distributions from which samples are drawn. Depending on the meta-distribution, the domain invariant methods could fail miserably, e.g., when label marginals of two joint distribution $P_{XY}$ and $Q_{XY}$ have non-overlapping support. Therefore, the authors do not make domain invariant assumption, but to operate in such a general setting, they instead use the information of marginal distribution, additional to the input sample data, to predict the label. Our setting is not as general, since our method works well when there exist regions on data space such that source and target domain are highly similar (there is a good hypothesis for both domains in that region). The mask network serves exactly that purpose, finding those regions (via masking feature). However, unlike Blanchard et al., we do not require a large set of source domains to work with.
>
> &nbsp;
>
> **Q2**: How does risk bound change with the number of available domains and number of examples in the training?
>
> Both Theorem 1 and 2 in our paper concern transferring from a single source domain to a target domain. In particular, Theorem 1 explains popular domain invariant methods, where a single hypothesis is learned to minimize source loss and target-source feature discrepancy. On the other hand, Theorem 2 suggests employing a set of hypotheses to classify both target and source domain, in which each hypothesis corresponds to a feature sub-space, and a specific region on the data space. Therefore, the message is that using a set of hypotheses improves transferring from a source domain to a target domain. In order to generalize this reasoning to transferring from a set of source domains to a target domain, we must first construct a single source mixture from the multiple source domains and then invoke Theorem 2. As such, the learned hypothesis set is dependent on the mixture weights and the number of source domains.
>
> &nbsp;
>
> **Q3**: Like in [1-2], can authors comment on the learning theoretic study?
>
> Our theory relates only to population losses and hence there isn't an approximation term involving sample complexity from learning theory. It is, therefore, more suitable to the scenario where samples are representative of the domains.
>
> &nbsp;
>
> **Q4**: As we have a large number of domains and a large number of training examples then one should be able to have optimal error or loss. I could not draw this conclusion from the theory that the authors provided. Can authors comment more on this?
>
> In order to transfer from multiple-source domains to a single target domain, we first construct a mixture, and the target loss is upper bounded by terms involving this source mixture and the target domain. Therefore, the asymptotic behavior of the bound as the number of domains increases is reflected through the convergence of the source mixture toward the mixture of all possible sources.
>
> We employ the hyper-parameters from original papers and the recent DomainBed library (Gulrajani & Lopez-Paz, 2021) for the baseline methods since they are carefully tuned.

---

> > ### Comment · Reviewer_crRo · 2021-11-26
> > **Comment on theoretical arguments**
> >
> > I thank the authors for their updated manuscript and changes in the paper. Some of my concerns regarding the literature and hyperparameters are resolved. I had more concerns regarding the theoretical arguments and these concerns are raised by other reviewers too.
> >
> > 1) "Our setting is not as general, since our method works well when there exist regions on data space such that source and target domain are highly similar (there is a good hypothesis for both domains in that region)."
> > Do we even need domain generalization if two domains are highly similar in the region?
> >
> > 2) " Theorem 1 explains popular domain invariant methods, where a single hypothesis is learned to minimize source loss and target-source feature discrepancy.",
> > I agree with other reviewers that this statement seems more of domain adaptation argument rather than domain generalization argument. Theoretical part for domain generalization seems to be weaker than what I earlier thought.
> >
> > Based on other reviews and comments I am tending to reduce my score from 8 to 6.

---

> > > ### Author Response · Authors · 2021-11-29
> > > **Response to Reviewer crRo (Theoretical arguments Q1)**
> > >
> > > We thank the reviewer for your feedback.
> > >
> > > **Q1**: *"Our setting is not as general, since our method works well when there exist regions on data space such that source and target domain are highly similar (there is a good hypothesis for both domains in that region).”* Do we even need domain generalization if two domains are highly similar in the region?"
> > >
> > > ---
> > >
> > > We believe any domain generalization needs some assumption about the relationship of source and unseen target domains to work well. We would like to further clarify the statement: *"Our method works well when there exist regions on data space such that source and target domain are highly similar".*
> > >
> > > By the above statement, we emphasize our LASSO works well if for an unseen target domain, we can extract target representations, each of which contains a group of label-informative features being similar to another group of label-informative features of source representations, hence the corresponding target representation is projected onto the same sub-space as these source representations. Moreover, since the sub-space contains label-informative features, we can preserve sufficient label information to learn a good hypothesis for this sub-space.
> > >
> > > Note that, according to our theory development, a sub-space index $m\in\\{0,1\\}^{D}$ represents a region on the original data space which has the same index $m$ i.e., $A_{m}=\\{ x:\Gamma(g(x))=m\\} \subset\mathcal{X}$.
> > > Therefore, projecting data to sub-spaces means that the data space is partitioned into regions i.e., $\mathcal{X} = \bigcup_{m=1}^{\mathcal{M}} A_{m}$, wherein each region, samples from source and target domains share the same characteristics (or are highly similar in terms of sub-space representation). That is why we stated: *"our method works well when there exist **regions on data space** such that source and target domain are highly similar."*
> > >
> > > In addition, by learning multiple hypotheses, each of which corresponds to a sub-space, LASSO allows sub-spaces to instance-wisely explore different groups of label-informative features, hence encouraging the diversity of latent representations for achieving better generalization ability. The reason is that given source examples with their labels, LASSO aims to explore possible compact groups of label-informative features which can predict the labels accurately. In the inference time, given a target example, if the feature extractor can successfully activate a group of label-informative features, this target example is projected and matched with corresponding source examples on a sub-space in which a good hypothesis is used to predict a label for the target example.
> > >
> > > Having said that, we believe our assumption of the appropriate decomposition of target representations to appropriate sub-spaces is sufficiently general. Finally, the qualified experimental results of LASSO on various datasets consolidate and confirm this claim.

---

> > > > ### Author Response · Authors · 2021-11-29
> > > > **Response to Reviewer crRo (Theoretical arguments Q2)**
> > > >
> > > > **Q2**: *"Theorem 1 explains popular domain invariant methods, where a single hypothesis is learned to minimize source loss and target-source feature discrepancy.”* I agree with other reviewers that this statement seems more of domain adaptation argument rather than domain generalization argument.  Theoretical  part  for  domain generalization seems to be weaker than what I earlier thought.
> > > >
> > > > ---
> > > >
> > > > The setting of domain generalization is highly similar to that of multiple source domain adaptation, except unseen versus seen target domains. Therefore, the theory developed for multiple source domain adaptation under the assumption of unseen target domains is applicable to domain generalization.
> > > >
> > > > In addition, the purpose of Theorem 1 is to see the factors influencing the performance on unseen target domains. From that, we motivate our multiple sub-spaces and hypotheses approach. It is worth noting that our Theorem 1 really tailors with the deep learning-based setting because the distribution shift gap is measured on a latent space, while the source and target losses engage the data space. We believe this is a non-trivial contribution because this explains why we can minimize the distribution gap on a latent space which expresses and follows exactly what we are doing when involving a deep learning-based feature extractor. To the best of our knowledge, existing works have established similar bounds, but these bounds engage the target/source losses and data shift on the original data space.
> > > >
> > > > Moreover, developing rigorous and impactful theories for sole domain generalization is still a challenging question to the community. The general assumption of unseen target domains generated from the same random process as source domains seems to be less informative and insufficiently robust to promote thorough theories with significantly practical impacts.
> > > >
> > > > Last but not least, to serve our LASSO, we develop Theorems 2 and 3 which shed light on the rationale of LASSO and explain how and why the sub-space indicator can learn label-informative sub-spaces which further helps to reduce the latent data shift on those sub-spaces.
> > > >
> > > > We realize that we cannot perfectly theoretically justify why projecting onto low-dimensional and label-informative sub-spaces assists in reducing the latent data shift. For this claim, we based on intuition, illustrative example (cf. Figure 1), and empirical experiments to verify. We hope this could be tolerable given the existence of many published works in our field and in domain generalization developed solely based on intuition without any supportive theory development.

---

### Official Review · Reviewer_43Db · 2021-11-03

**Correctness:** 3
**Technical Novelty And Significance:** 3
**Empirical Novelty And Significance:** 3
**Recommendation:** 6
**Confidence:** 4

**Main Review:**

- Strength
  - The proposed method is solid and novel.
  - The empirical results show solid improvement over baselines with a large network backbone.

- Weakness
  - Theoretical arguments are not particularly strong. For example, Theorem 1 and 2 are rather for domain adaptation when one has access to the target domain, so it is unclear how it is relevant to the domain generalization context. While formulating a bound with latent space might be new, but it does not seem to add any better understanding for domain generalization.
  - \Gamma is optimized based on Equation (7) and it is unclear what authors mean by "our principle is to encourage \Gamma_{d} to becomes more independent". Are there any part of the optimization problem that promotes independence?
  - Might be good to add a baseline without updating \Gamma (i.e., adding a dropout layer).

- Misc
  - It is unclear what it means by "but the gains shrink with ResNet50 since larger ResNet backbones are known to generalize better" in Section 4.1.3.
  - The trend observed in the paper on the performance w.r.t. the network size seems interesting. Might be good to add more experimental results with deeper networks to see if trend holds true.

**Summary Of The Paper:**

The paper presents LASSO, a Latent Sub-Spaces Orientation to tackle domain generalization problem. The paper involves both theoretical and empirical results.

Theorem 1 says the target domain loss is bounded by terms involving source domain losses, label shift, and the latent data shift (which reminds me of the theorems from the seminal work by Ben-David et al. for domain adaptation). Theorem 2 extends Theorem 1 but with latent subspaces. Theorem 3 says that for a given subspace indicator \Gamma, the mutual information between the sub-space and label is lower bounded by the sum of negative source domain losses.

The proposed method essentially introduces the subspace-indicating binary variables in the neural network. This seems to work similarly to the dropout, but with more structure and with learned and data-dependent drop rates. Inference is also done by generating multiple subspace-indicating binary masks and ensemble them.

Experimental results are provided on standard DG benchmarks. The trend seems clear that for shallow network (e.g., AlexNet, ResNet-18) the performance is not necessarily better than previous methods, but with deeper network (e.g., ResNet-50) the improvement is  somewhat significant.

**Summary Of The Review:**

While there is a question on the significance of the theoretical results, overall the paper proposed a new domain generalization algorithm that is both technically and empirically solid.

---

> ### Author Response · Authors · 2021-11-19
> **Response to Reviewer 43Db (1/2)**
>
> Thank you very much for your constructive comments. We hope that we can address some of your points as presented below.
>
> &nbsp;
>
> ### **Theorem 1 and Theorem 2**
> We agree with the reviewer that Theorem 1 and Theorem 2 would be more straightforward in the context of domain adaptation. However, these theorems can be used to demonstrate limitations of current methods in DG context and point out the direction which we can employ to address DG problems.
>
> For example, since the target domain is unknown beforehand when training, a large number of works propose learning domain-invariant features on a full high-dimensional latent space together with a single hypothesis on top of these domain-invariant features. Nonetheless, due to the great variance of unseen target distributions on the full high-dimensional latent space, the latent data shift is possibly high in many cases, which hurts the generalization ability of the single hypothesis on unseen target domains.
>
> Evidently, given a latent representation $z$ with a label $y$ in the high-dimensional latent space, only a small portion of its features known as *label-informative features* is highly relevant to the label $y$, while the remaining ones are redundant. This observation is supported by our ablation study about the effect of $\tau$ which is presented in Table 4 (main paper).
>
> By eliminating irrelevant features and grouping the latent representations of data examples across multiple domains with the same set of label-informative features, we can form latent label-informative sub-spaces to reduce the latent data shift, whereas preserving *sufficient label-information* for training *good hypotheses* on those *latent sub-spaces*.  This can be explained by the fact that unseen target and source examples with the same label-informative features are projected onto the same latent sub-space on which the latent data shift between the mixture of source domains and target domain becomes smaller due to the compactness of this sub-space compared with the full high-dimensional latent space.
> From our ablation study about sub-space representation, it can be seen that the latent is more compact and easy to classify on sub-space. The numerical results also demonstrate that the latent data shift can be reduced on sup-space.
>
> By learning multiple hypotheses, each of which corresponds to a sub-space, LASSO allows sub-spaces to instance-wisely explore different groups of label-informative features, hence encouraging the diversity of latent representations for achieving better generalization ability, which concurs with the principle in (Huang et al., 2020; Chattopadhyay et al., 2020; Blanchard et al., 2021).
>
> The reason is that given source examples with their labels, LASSO aims to explore possible compact groups of label-informative features which can predict accurately the labels. In the inference time, given a target example, if the feature extractor can successfully activate a group of label-informative features, this target example is projected and matched with corresponding source examples on a sub-space in which a good hypothesis is used to predict a label for the target example.
>
> **Update:** Following the suggestion of the reviewer, we have revised the discussion after Theorem 1 to further clarify motivation and added Figure 1 with a brief description to better illustrate the overall idea.

---

> > ### Author Response · Authors · 2021-11-19
> > **Response to Reviewer 43Db (2/2)**
> >
> > ### **Attribute-based $\Gamma$**
> >
> > Independence promotion is based on the design of Attribute-based $\Gamma$ instead of the optimization in Eq. (7). More specifically, we consider that a feature vector $z$ would consist of attributes e.g.
> > $z=\left [z_{1},...,z_{D}\right ]\in\mathbb{R}^{D'\times D}$ whereas each $z_{d}\in\mathbb{R}^{D'}$ known as **an attribute**
> > consists of $D'$ features and model sub-space indicator as $\Gamma\left(z\right)=\left[\Gamma_{1}\left(z_{1}\right),...,\Gamma_{D}\left(z_{D}\right)\right]\in\left[0,1\right]^{D}$. Consequently, $\Gamma_{d}\left(z\right)=\Gamma_{d}\left(z_{d}\right)$ is computed based solely on the group of $D'$ features in the attribute
> > $z_{d}$ itself rather than full latent $z$. That means attribute-based $\Gamma_d$ depends on attributes which are shared across domains instead of domains, hence, becomes more independent from domain information. Moreover, by Theorem 3, model can still learn meaningful attributes as long as the performance on source domains is guaranteed. In unseen domains, the target of sub-space indicator $\Gamma$ is to detect label-informative attributes which is learned in source domains instead of identifying all label-informative attributes.
> >
> > Looking at each $\Gamma_{d}\left(z_{d}\right)$ in detail, since $z_d$ come from different source domains, $\Gamma_d$ play as a domain invariant hypothesis which output {0, 1  } identifying label-informative attributes across source domains. Therefore, in target domain, $\Gamma_{d}\left(z_{d}\right)$ only predict $z_d$ is label-informative attribute **which are learned in source domains** or not.
> >
> > Our intuition is supported by ablation study about the effect of the attribute-based indicator $\Gamma_d$ which is presented in Table 5, Appendix B.1.
> >
> > &nbsp;
> >
> > ### **Discussion and comparison with Random Dropout**
> >
> > We would like to report additional experimental results in comparison with "Random Dropout" Baseline (Srivastava et al., 2014) for different dropping rate as follows:
> >
> >
> > ***Table 1: Classification Accuracy on PACS with ResNet18***
> >
> > | Feature Dropping Rate | Backbone  | Photo | Art-painting | Cartoon | Sketch | Average |
> > | :---- | :----: | :----: | :----: | :----: | :----: |----: |
> > 0\% | ResNet18 | 96.17 +/- 0.27 | 78.42 +/- 1.29 | 76.73 +/- 0.64 | 75.40 +/- 0.37 | 81.68 |
> > 10\% | ResNet18 | 96.29 +/- 0.66 | 81.01 +/- 0.33 | 76.98 +/- 0.67 | 75.60 +/- 0.81 | 82.47 |
> > 30\% | ResNet18 | 96.41 +/- 0.60 | 79.73 +/- 0.69 | 77.68 +/- 0.28 | 75.22 +/- 1.07 | 82.26 |
> > 50\% | ResNet18 | 96.59 +/- 0.14 | 81.20 +/- 0.02 | 77.92 +/- 0.24 | 75.98+/-0.17 | 82.92 |
> > 70\% | ResNet18 | 96.29 +/- 0.14 | 79.93 +/- 0.72 | 78.26 +/- 0.67 | 74.89 +/- 0.09 | 82.34
> > LASSO-$\tau$ | ResNet18 | 94.76 +/- 0.03 | 82.17 +/- 0.60 | 78.37 +/- 0.55 | 77.34 +/- 0.37 | **83.16** |
> > LASSO-En | ResNet18 | 94.76 +/- 0.03 | 82.02 +/- 0.75 | 77.65 +/- 0.76 | 77.07 +/- 0.42 | 82.87 |
> >
> > &nbsp;
> >
> > ***Table 2: Classification Accuracy on PACS with ResNet50***
> >
> > | Feature Dropping Rate | Backbone  | Photo | Art-painting | Cartoon | Sketch | Average |
> > | :---- | :----: | :---- | :---- | :---- | :---- |----: |
> > 0\% | ResNet50 | 97.20 +/- 0.30  | 84.70 +/- 0.40 | 80.80 +/- 0.60 | 79.30 +/- 1.00 | 85.50 |
> > 10\% | ResNet50 | 97.96 +/- 0.26 | 86.96 +/- 0.37  | 78.92 +/- 1.44 | 79.96 +/- 0.64  | 85.95 |
> > 30\% | ResNet50 | 98.08 +/- 0.32 | 87.26 +/- 0.63 | 80.11 +/- 1.10 | 81.31 +/- 0.80 | 86.69 |
> > 50\% | ResNet50 | 97.60 +/- 0.04 | 87.16 +/- 0.13 | 80.46 +/- 0.24 | 80.80 +/- 0.77 | 86.51 |
> > 70\% | ResNet50 | 97.08 +/- 0.47 | 86.33 +/- 0.73 | 81.78 +/- 0.98 | 80.57 +/- 1.01 | 86.44 |
> > LASSO-$\tau$ | ResNet50 | 96.62 +/- 0.56 | 87.81 +/- 0.69 | 82.99 +/- 0.71 | 82.33 +/- 0.20 | **87.43** |
> > LASSO-En | ResNet50 | 96.93 +/- 0.65 | 87.23 +/- 0.57 | 82.55 +/- 1.09 | 81.95 +/- 0.37 | 87.15 |
> >
> > The results in Table 1 and Table 2 indicate that ERM with an appropriate dropping rate (i.e., 50\% for ResNet18 and 30\% for ResNet50) is able to achieve comparative performance to most current baselines.
> >
> > In fact, "Dropout" has a quite similar effect to the representation as LASSO, which increases the diversity of latent representations. However, LASSO goes one step further compared to "Dropout". Specifically, representation is used as it is during inference time, without making the use of information in the target data, e.g., using the full network for inference, or average over a set of randomly "Dropout" networks (MC dropout).
> > On the contrary, the sub-space indicator in LASSO additionally takes into account the information from the target domain to select the suitable masked network which is likely to generalize better to this particular target domain. This is the motivation behind the selection of appropriate sub-space.
> >
> > We have added additional experiments about "Random Dropout" baseline to the Appendix B.3.

---

> > > ### Author Response · Authors · 2021-11-29
> > > **Looking forward to Reviewer's feedback**
> > >
> > > Dear Reviewer,
> > >
> > > Thanks again for your valuable comments and reviews. The discussion period is coming to a close, and we will not be able to directly engage in further discussions after that point. We would really appreciate a reply as to whether our response and clarifications have addressed the issues raised in your review or anything else we can address.
> > >
> > > Authors

---

> > > > ### Comment · Reviewer_43Db · 2021-11-30
> > > > **Thanks for the response**
> > > >
> > > > Thanks authors for their response.
> > > >
> > > > Thanks for clarification between relations of Theorem 1, 2 and domain generalization. I think it is safe to say that the proposed DG methods are not direct derivatives of the theorems, but more of a high-level motivation. As other reviewer also commented, I would be also curious whether theorem 2 shows better bound than theorem 1 as they are used to motivate the proposed DG method, but as authors commented it does not seem matter. This point should be clarified in the paper.
> > > >
> > > > Thanks for running additional experiments with random dropout. It seems having structures as in the proposed method helps over random dropout, though the performance gain is marginal.
> > > >
> > > > Overall I still think the paper is in a good shape with a solid technical contribution and empirical verification, though theorems are not as directly connected to the proposed algorithm, as other reviewers also pointed out. I would keep my rating.

---

### Author Response · Authors · 2021-11-22
**Summary of Revision**

We thank all Reviewers for their insightful reviews and supportive suggestions. We hope our response addresses your concerns and we upload the revised manuscript with a few modifications, detailed as follows:

The following updates have been incorporated in the **main manuscript**:
- Revise the discussion after Theorem 1 to further clarify motivation and add Figure 1 with a brief description to better illustrate the overall idea.
- Revise the discussion about how attributed-based $\Gamma$ help better generalization in Section 3.4.1.
- Add some works published in 2021 to the benchmarks (Table 1, 2, 3)

The following updates have been inserted in the **Appendix**.
- Move the ablation study about attributed-based $\Gamma$ to the Appendix B.1.
- Add discussion and additional experiments about "Sampling" for model training to the Appendix B.2.
- Add discussion and additional experiments about "Random Dropout" baseline to the Appendix B.3.
- Move the "evaluation protocol" section to the Appendix C.2.

We would be more than happy to discuss any further questions!

---

### Author Response · Authors · 2021-12-03
**The connection between our theory and proposed method**

We really appreciate the reviewers for their great efforts and active discussions of our paper. According to the discussions, the only concern left is the connection of our proposed theory and practical method.  Although our theory is not perfect because we need to make an assumption based on the intuition which can be only empirically verified in our ablation studies,  we believe the theory is helpful and supportive to our practical method.

For the theoretical part, to the best of our knowledge, unlike domain adaptation, there is currently no rigorous theory for domain generalization to tightly bound the loss on unseen target domains due to the great variance of unseen target domains. The general assumption of unseen target domains generated from the same random process (i.e., a distribution over distribution) as source domains seems to be overly general and insufficient to motivate more rigorous theory. Most of work in domain generalization needs to make some assumption to go further.

Moreover, many works aim to learn domain-invariant features on source domains. From conceptual viewpoint, these works simply borrow the principle of learning domain-invariant features from domain adaptation, but did not mention to how the domain-invariant features generalize to unseen target domains or how to quantify the target loss over unseen target domains caused by the domain-invariant features on source domains.

Comparing to these works, with the assistance of Theorem 1 in our work, we can realize the factors influencing the loss on unseen target domains. We regret that to motivate from *single-space & single hypothesis* to *multiple-spaces & multiple-hypotheses*, we need to base on the intuition and motivation that when grouping data examples by label-informative features and projecting them to these sub-spaces, we can reduce the domain shift, while preserving sufficient label information to learn good sub-space hypotheses.


Actually, this can be partly confirmed from the monotonicity property of a general $f-divergence$ for which we have: $D_f(P_{X,Y} || Q_{X,Y}) \geq D_f(P_{X} || Q_{X})$, implying that projecting to a sub-space helps to reduce $f-divergence$.

However, if we cannot choose a good set of features $X$, we cannot train good and qualified hypothesis on this sub-space. Our Theorem 3 links to information theory to state: by minimizing the losses of hypotheses on sub-spaces, we can maximize the mutual information $I(\Gamma(g(X)) \odot X, g(X), Y)$ to enable the sub-space indicator $\Gamma$ to learn good and label-informative features.

---

### Decision · Program_Chairs · 2022-01-20

**Decision:**

Reject

**Comment:**

This paper proposes a novel method for improving domain generalization based on the idea of learning different subspaces for each domain. Authors provide theoretical analysis related to their proposal and further evaluate their proposed method on a subset of DomainBed benchmark.


**Strong Points:**

- The paper is well-written.

- The proposed method is novel.

- Authors provide theoretical analysis in support of their proposal.

- The theoretical results seem to be correct.

- Empirical evaluation shows that the proposed method improves over baselines on a subset of datasets included in the DomainBed benchmark.

**Weak Points:**

- The complexity of the theoretical results makes it very difficult for the reader to get any intuition about the underlying mechanisms at play.

- The theoretical analysis is disconnected from the proposed algorithm. It is hard to see how one could end up proposing such an algorithm following the theoretical results. I suggest that authors would consider reorganizing the paper with less emphasis on the theoretical part, perhaps simplifying the theoretical results and pushing the rest to appendix.

- The empirical evaluation can be improved significantly. Domain generalization is a very well-established area at this point. WILDS is a carefully designed and well-known benchmark and showing improvement in that benchmark would be very convincing but unfortunately authors do not discuss or even refer to it. They instead report their results on a subset of datasets used in DomainBed benchmark. The DomainBed benchmark is less challenging than WILDS but even following DomainBed closely and reporting the 3 evaluation metrics on all 7 datasets would have been satisfying. However, authors only report the results on 3 datasets. Reporting the results on a diverse group of datasets is particularly important in the case of Domain Generalization because we know that many methods are able to show improvements on a few datasets but it is challenging to beat the baselines on a significant majority of datasets.

**Final Decision Rationale**:

This is a borderline paper. On one hand, the proposed method is interesting and novel. On the other hand, the theoretical contributions are very limited and the empirical evaluation is not strong enough for acceptance. Given that all weak points mentioned above can be addressed, I recommend rejection and I sincerely hope that authors would strengthen their paper by addressing them before resubmitting their work.